# Entropy-based adaptive Hamiltonian Monte Carlo

**Marcel Hirt**
Department of Statistical Science
University College London, UK
`marcel.hirt.16@ucl.ac.uk`

**Michalis K. Titsias**
DeepMind
London, UK
`mtitsias@google.com`

**Petros Dellaportas**
Department of Statistical Science
University College London, UK
Department of Statistics
Athens Univ. of Economics and Business, Greece
and The Alan Turing Institute, UK

## Abstract

Hamiltonian Monte Carlo (HMC) is a popular Markov Chain Monte Carlo (MCMC) algorithm to sample from an unnormalized probability distribution. A leapfrog integrator is commonly used to implement HMC in practice, but its performance can be sensitive to the choice of mass matrix used therein. We develop a gradient-based algorithm that allows for the adaptation of the mass matrix by encouraging the leapfrog integrator to have high acceptance rates while also exploring all dimensions jointly. In contrast to previous work that adapt the hyper-parameters of HMC using some form of expected squared jumping distance, the adaptation strategy suggested here aims to increase sampling efficiency by maximizing an approximation of the proposal entropy. We illustrate that using multiple gradients in the HMC proposal can be beneficial compared to a single gradient-step in Metropolis-adjusted Langevin proposals. Empirical evidence suggests that the adaptation method can outperform different versions of HMC schemes by adjusting the mass matrix to the geometry of the target distribution and by providing some control on the integration time.

## 1 Introduction

Consider the problem of sampling from a target density $\pi$ on $\mathbb{R}^d$ of the form $\pi(q) \propto e^{-U(q)}$, with a *potential* energy $U \colon \mathbb{R}^d \to \mathbb{R}$ being twice continuously differentiable. HMC methods [20, 46, 9] sample from a *Boltzmann-Gibbs* distribution $\mu(q,p) \propto e^{-H(q,p)}$ on the phase-space $\mathbb{R}^{2d}$ based on the (separable) *Hamiltonian* function

$$H(q,p) = U(q) + K(p) \quad \text{with} \quad K(p) = \frac{1}{2}p^\top M^{-1}p.$$

The Hamiltonian represents the total energy that is split into a potential energy term $U$ and a *kinetic* energy $K$ which we assume is Gaussian for some symmetric positive definite *mass matrix* $M$. Suppose that $(q(t), p(t))_{t \in \mathbb{R}}$ evolve according to the differential equations

$$\frac{\mathrm{d}q(t)}{\mathrm{d}t} = \frac{\partial H(q(t), p(t))}{\partial p} = M^{-1}p(t) \quad \text{and} \quad \frac{\mathrm{d}p(t)}{\mathrm{d}t} = -\frac{\partial H(q(t), p(t))}{\partial q} = -\nabla U(q(t)). \quad (1)$$

Let $(\varphi_t)_{t \geqslant 0}$ denote the flow of the Hamiltonian system, that is for fixed $t$, $\varphi_t$ maps each $(q,p)$ to the solution of (1) that takes value $(q,p)$ at time $t = 0$. The exact HMC flow $\varphi$ preserves

35th Conference on Neural Information Processing Systems (NeurIPS 2021).

volume and conserves the total energy *i.e.* $H \circ \varphi_t = H$. Consequently, the Boltzmann-Gibbs distribution $\mu$ is invariant under the Hamiltonian flow, that is $\mu(\varphi_t(E)) = \mu(E)$ for any Borel set $E \subset \mathbb{R}^{2d}$. Furthermore, the flow satisfies the generalized *reversibility* condition $\mathcal{F} \circ \varphi_t = \varphi_{-t} \circ \mathcal{F}$ with the flip operator $\mathcal{F}(q, p) = (q, -p)$. Put differently, the Hamiltonian dynamics go backward in time by negating the velocity. If an analytical expression for the exact flow were available, one could sample from $\mu$ using the invariant Markov chain that at state $(q, p)$ first draws a new velocity $p' \sim \mathcal{N}(0, M)$ with the next state set to $\varphi_T(q, p')$ for some *integration time* $T > 0$. Such a velocity refreshment is necessary as the HMC dynamics preserve the energy and so cannot be ergodic. However, the Hamiltonian flow cannot be computed exactly, except for very special potential functions. Numerical approximations to the exact solution of Hamiltonian's equations are thus routinely used, most commonly the *leapfrog* method, also known as (velocity) Verlet integrator [28, 10]. For a step size $h > 0$ and $L$ steps, such an algorithm updates the previous state $q_0$ and a new velocity $p_0 \sim \mathcal{N}(0, M)$ by setting, for $0 \leqslant \ell \leqslant L - 1$,

$$p_{\ell + \frac{1}{2}} = p_\ell - \frac{h}{2} \nabla U(q_\ell); \quad q_{\ell+1} = q_\ell + h M^{-1} p_{\ell + \frac{1}{2}}; \quad p_{\ell+1} = p_{\ell + \frac{1}{2}} - \frac{h}{2} \nabla U(q_{\ell+1}). \quad (2)$$

This scheme can be motivated by splitting the Hamiltonian wherein the kick mappings in the first and third step update only the momentum, while the drift mapping in the second step advances only the position $q$ with constant speed. For $T = Lh$, the leapfrog integrator approximates $\varphi_T(q_0, p_0)$ by $(q_L, p_L)$ while also preserving some geometric properties of $\varphi$, namely volume preservation and generalized reversibility. The leapfrog method is a second-order integrator, making an $\mathcal{O}(h^2)$ energy error $H(q_L, p_L) - H(q_0, p_0)$. A $\mu$-invariant Markov chain can be constructed using a Metropolis-Hastings acceptance step. More concretely, the proposed state $(q_L, p_L)$ is accepted with the acceptance rate $a(q_0, p_0) = \min\{1, \exp\left[-\left(H(q_L, p_L) - H(q_0, p_0)\right)\right]\}$, while the next state is set to $\mathcal{F}(q_0, p_0)$ in case of rejection, although the velocity flip is inconsequential for full refreshment strategies.

We want to explore here further the generalised speed measure introduced in [54] for adapting RWM or MALA that aim to achieve fast convergence by constructing proposals that (i) have a high average log-acceptance rate and (ii) have a high entropy. Whereas the entropy of the proposal in RWM or MALA algorithms can be evaluated efficiently, the multi-step nature of the HMC trajectories makes this computation less tractable. The recent work in [41] consider the same adaptation objective by learning a normalising flow that is inspired by a leapfrog proposal with a more tractable entropy by masking components in a leapfrog-style update via an affine coupling layer as used for RealNVPs [19]. [60] sets the integration time by maximizing the proposal entropy for the exact HMC flow in Gaussian targets, while choosing the mass matrix to be the inverse of the sample covariance matrix.

## 2  Related work

The choice of the hyperparameters $h, L$ and $M$ can have a large impact on the efficiency of the sampler. For fixed $L$ and $M$, a popular approach for adapting $h$ is to target an acceptance rate of around 0.65 which is optimal for iid Gaussian targets in the limit $d \to \infty$ [8] for a given integration time. HMC hyperparameters have been tuned using some form of *expected squared jumping distance* (ESJD) [49], using for instance Bayesian optimization [56] or a gradient-based approach [40]. A popular approach suggested in [32] tunes $L$ based on the ESJD by doubling $L$ until the path makes a U-turn and retraces back towards the starting point, that is by stopping to increase $L$ when the distance to the proposed state reaches a stationary point [4]; see also [57] for a variation and [48] for a version using sequential proposals. Modern probabilistic programming languages such as Stan [12], PyMC3 [51], Turing [23, 58] or TFP [39] furthermore allow for an adaptation of a diagonal or dense mass-matrix within NUTS based on the sample covariance matrix. The Riemann manifold HMC algorithm from [25] has been suggested that uses a position dependent mass matrix $M(x)$ based on a non-separable Hamiltonian, but can be computationally expensive, requiring $\mathcal{O}(d^3)$ operations in general. An alternative to choose $M$ or more generally the kinetic energy $K$ was proposed in [43] by analysing the behaviour of $x \mapsto \nabla K(\nabla U(x))$. Different pre-conditioning approaches have been compared for Gaussian targets in [38]. A popular route has also been to first transform the target using tools from variational inference as in [31] and then run a HMC sampler with unit mass matrix on the transformed density with a more favourable geometry.

A common setting to study the convergence of HMC assumes a log-concave target. In the case that $U$ is $m_1$-strongly convex and $m_2$-smooth, [45, 15] analyse the ideal HMC algorithm with unit mass matrix where a higher condition number $\kappa = m_2/m_1$ implies slower mixing: The relaxation time, *i.e.* the inverse of the spectral gap, grows linear in $\kappa$, assuming the integration time is set to $T = \frac{1}{2\sqrt{m_2}}$. [14] establish non-asymptotic upper bounds on the mixing time using a leap-frog integrator where the step size $h$ and the number $L$ of steps depends explicitly on $m_1$ and $m_2$. Convergence guarantees are established using conductance profiles by obtaining (i) a high probability lower bound on the acceptance rate and (ii) an overlap bound, that is a lower bound on the KL-divergence between the HMC proposal densities at the starting positions $q_0$ and $q_0'$, whenever $q_0$ is close to $q_0'$. While such bounds for controlling the mixing time might share some similarity with the generalised speed measure (GSM) considered here, they do not lend themselves easily to a gradient-based adaptation.

## 3 Entropy-based adaptation scheme

We derive a novel method to approximate the entropy of the proposed position after $L$ leapfrog steps. Our approximation is based on the assumption that the Hessian of the target is locally constant around the mid-point of the HMC trajectory, which allows for a stochastic trace estimator of the marginal proposal entropy. We develop a penalised loss function that can be minimized using stochastic gradient descent while sampling from the Markov chain in order to optimize a GSM.

### 3.1 Marginal proposal entropy

Suppose that $CC^\top = M^{-1}$, where $C$ is defined by some parameters $\theta$ and can be a diagonal matrix, a full Cholesky factor, etc. Without loss of generality, the step size $h > 0$ can be fixed. We can reparameterize the momentum resampling step $p_0 \sim \mathcal{N}(0, M)$ by sampling $v \sim \mathcal{N}(0, \mathrm{I})$ and setting $p_0 = C^{-\top}v$. One can show by induction, cf. Appendix E for details, that the $L$-th step position $q_L$ and momentum $p_L$ of the leapfrog integrator can be represented as a function of $v$ via

$$q_L = \mathcal{T}_L(v) = q_0 - \frac{Lh^2}{2}M^{-1}\nabla U(q_0) + LhCv - h^2 M^{-1}\Xi_L(v), \tag{3}$$

and

$$p_L = \mathcal{W}_L(v) = C^{-\top}v - \frac{h}{2}\left[\nabla U(q_0) + \nabla U \circ \mathcal{T}_L(v)\right] - h\sum_{i=1}^{L-1}\nabla U \circ \mathcal{T}_i(v) \tag{4}$$

where

$$\Xi_L(v) = \sum_{i=1}^{L-1}(L-i)\nabla U \circ \mathcal{T}_i(v), \tag{5}$$

see also [42, 21, 14] for the special case with an identity mass matrix. Observe that for $L = 1$ leap-frog steps, this reduces to a MALA proposal with preconditioning matrix $M^{-1}$.

Under regularity conditions, see for instance [21], the transformation $\mathcal{T}_L \colon \mathbb{R}^d \to \mathbb{R}^d$ is a C$^1$-diffeomorphism. With $\nu$ denoting the standard Gaussian density, the density $r_L$ of the HMC proposal for the position $q_L$ after $L$ leapfrog steps is the pushforward density of $\nu$ via the map $\mathcal{T}_L$ so that[1]

$$\log r_L(\mathcal{T}_L(v)) = \log \nu(v) - \log|\det \mathsf{D}\mathcal{T}_L(v)|. \tag{6}$$

Observe that the density depends on the Jacobian of the transformation $\mathcal{T}_L \colon v \mapsto q_L$. We would like to avoid computing $\log|\det \mathsf{D}\mathcal{T}_L(v)|$ exactly. Define the residual transformation

$$\mathcal{S}_L \colon \mathbb{R}^d \to \mathbb{R}^d, \; v \mapsto \frac{1}{Lh}C^{-1}\mathcal{T}_L(v) - v. \tag{7}$$

Then $\mathsf{D}\mathcal{T}_L(v) = LhC(\mathrm{I}+\mathsf{D}\mathcal{S}_L(v))$ and consequently

$$\log|\det \mathsf{D}\mathcal{T}_L(v)| = d\log(Lh) + \log|\det C| + \log|\det(\mathrm{I}+\mathsf{D}\mathcal{S}_L(v))|. \tag{8}$$

Combining (6) and (8) yields the log-probability of the HMC proposal

$$\log r_L(\mathcal{T}_L(v)) = \log \nu(v) - d\log(Lh) - \log|\det C| - \log|\det(\mathrm{I}+\mathsf{D}\mathcal{S}_L(v))|. \tag{9}$$

---

[1]We denote the Jacobian matrix of a function $f\colon \mathbb{R}^d \to \mathbb{R}^d$ at the point $x$ as $\mathsf{D}f(x)$.

Comparing the equations (3) and (7), one sees that $\mathcal{S}_L(v) = c - \frac{h}{L}C^\top \Xi_L(v)$ for some constant $c \in \mathbb{R}^d$ that depends on $\theta$ but is independent of $v$ and consequently, $\mathsf{D}\mathcal{S}_L(v) = -\frac{h}{L}C^\top \mathsf{D}\Xi_L(v)$. We next show a recursive expression for $\mathsf{D}\mathcal{S}_L$ with a proof given in Appendix B.

**Lemma 1** (Jacobian representation). *It holds that* $\mathsf{D}\mathcal{S}_1 = 0$ *and for any* $\ell \in \{2, \dots, L\}$, $v \in \mathbb{R}^d$,

$$\mathsf{D}\mathcal{S}_\ell(v) = -h^2 \sum_{i=1}^{\ell-1} (\ell - i)\frac{i}{\ell} C^\top \nabla^2 U\left(\mathcal{T}_i(v)\right) C\left(\mathrm{I} + \mathsf{D}\mathcal{S}_i(v)\right). \tag{10}$$

*In particular,* $\mathsf{D}\mathcal{S}_\ell(v)$ *is a symmetric matrix. Suppose further that* $L^2 h^2 < \sup_{q \in \mathbb{R}^d} \frac{1}{4\|C^\top \nabla^2 U(q)C\|_2}$. *Then for any* $\ell \in \{1, \dots, L\}$ *and* $v \in \mathbb{R}^d$, *we have* $\|\mathsf{D}\mathcal{S}_\ell(v)\|_2 < \frac{1}{8}$.

Notice that the recursive formula (10) requires computing $\frac{1}{2}L(L-1)$ terms, each involving the Hessian, in order to compute the Jacobian after $L$ leapfrog steps. Consider for the moment a Gaussian target with potential function $U(q) = \frac{1}{2}(q - q_\star)^\top \Sigma^{-1}(q - q_\star)$ for $q_\star \in \mathbb{R}^d$ and positive definite $\Sigma \in \mathbb{R}^{d \times d}$. Then, due to (10), for any $q \in \mathbb{R}^d$, $v \in \mathbb{R}^d$,

$$\mathsf{D}\mathcal{S}_L(v) = -h^2 \sum_{i=1}^{L-1} (L - i)\frac{i}{L} C^\top \Sigma^{-1} C(\mathrm{I} + \mathsf{D}\mathcal{S}_i(v)) = D_L + R_L(v),$$

where

$$D_L = -h^2 C^\top \Sigma^{-1} C\left(\sum_{i=1}^{L-1}(L-i)\frac{i}{L}\right) = -h^2 \frac{L^2 - 1}{6} C^\top \Sigma^{-1} C \tag{11}$$

and a remainder term $R_L(v) = -h^2 C^\top \Sigma^{-1} C\left(\sum_{i=1}^{L-1}(L-i)\frac{i}{L}\mathsf{D}\mathcal{S}_i(v)\right)$. From Lemma 1, we see that if $\|C^\top \Sigma^{-1} C\|_2 \leqslant \frac{1}{4h^2 L^2}$, then $\mathrm{I} + \mathsf{D}\mathcal{S}_L(v)$ and $-\mathsf{D}\mathcal{S}_L(v)$ are positive definite. Then $R_L$ is also positive definite and $\log \det(\mathrm{I} + D_L) \leqslant \log |\det(\mathrm{I} + \mathsf{D}\mathcal{S}_L(v))|$ and we can maximize the lower bound instead. Put differently, for Gaussian targets, $\mathsf{D}\mathcal{S}_L$ can be decomposed into a component $D_L$ that contains all terms that are linear in $h^2 C^\top \Sigma^{-1} C$ and that does not require a recursion; plus a component $R_L$ that contains terms that are higher than linear in $h^2 C^\top \Sigma^{-1} C$ and that needs to be solved recursively. Our suggestion is to ignore this second term. Note that $R_2 = 0$ and an extension can be to include higher order terms $\mathcal{O}\left(\left[h^2 C^\top \Sigma^{-1} C\right]^k\right)$, $k > 1$, in the approximation $D_L$.

For an arbitrary potential energy $U$, equation (10) shows that evaluating $\mathsf{D}\mathcal{S}_L$ leads to a non-linear function of the Hessians evaluated along the different points of the leapfrog-trajectory. We suggest to replace it with a first order term with one Hessian evaluation which is however scaled accordingly. Concretely, we maximize

$$\mathcal{L}(\theta) = \log|\det(\mathrm{I} + D_L)| \quad \text{with} \quad D_L = -h^2 \frac{L^2 - 1}{6} C^\top \nabla^2 U(q_{\lfloor L/2 \rfloor})C \tag{12}$$

as an approximation of $\log|\det(\mathrm{I} + \mathsf{D}\mathcal{S}_L)|$. The intuition is that we assume that the target density can be approximated locally by a Gaussian one with precision matrix $\Sigma^{-1}$ in (11) given by the Hessian of $U$ at the mid-point $q_{\lfloor L/2 \rfloor}$ of the trajectory. We want to optimize $\mathcal{L}(\theta)$ given in (12) even if we do not have access to the Hessian $\nabla^2 U$ explicitly, but only through Hessian-vector products $\nabla^2 U(q)w$ for some vector $w \in \mathbb{R}^d$. Vector-Jacobian products $\mathtt{vjp}(f, x, w) = w^\top \mathsf{D}f(x)$ for differentiable $f\colon \mathbb{R}^d \to \mathbb{R}^d$ can be computed efficiently via reverse-mode automatic differentiation, so that $\nabla^2 U(q)w = \mathtt{vjp}(\nabla U, q, w)^\top$ can be evaluated with complexity linear in $d$.

Suppose the multiplication with $D_L$ is a contraction so that all eigenvalues of $D_L$ have absolute values smaller than one. Then one can apply a Hutchinson stochastic trace estimator of $\log|\det(\mathrm{I}_d + D_{,L})|$ with a Taylor approximation, truncated and re-weighted using a Russian-roulette estimator [44], see also [29, 5, 13] for similar approaches in different settings. More concretely, let $N$ be a positive random variable with support on $\mathbb{N}$ and let $p_k = \mathbb{P}(N \geqslant k)$. Then,

$$\mathcal{L}(\theta) = \log \det(\mathrm{I} + D_L) = \mathbb{E}_{N,\varepsilon}\left[\sum_{k=1}^{N} \frac{(-1)^{k+1}}{kp_k}\varepsilon^\top (D_L)^k \varepsilon\right], \tag{13}$$

where $\varepsilon$ is drawn from a Rademacher distribution. While this yields an unbiased estimator for $\mathcal{L}(\theta)$ and its gradient as shown in Appendix A.1 if $D_L$ is contractive, it can be computationally expensive if $N$ has a large mean or have a high variance if $D_L$ has an eigenvalue that is close to 1 or $-1$, see [44, 17]. Since both the first order Gaussian approximation as well as the Russian Roulette estimator hinges on $D_L$ having small absolute eigenvalues, we consider a constrained optimisation approach that penalises such large eigenvalues. For the random variable $N$ that determines the truncation level in the Taylor series, we compute $b_N = (D_L)^N \varepsilon / \|(D_L)^N \varepsilon\|_2$ and $\mu_N = b_N^\top D_L b_N$. Note that this corresponds to applying $N$ times the power iteration algorithm and with $|\lambda_1| > |\lambda_2| \geqslant \ldots \geqslant |\lambda_d|$ denoting the eigenvalues of the symmetric matrix $D_L$, almost surely $\mu_n \to \lambda_1$ for $n \to \infty$, see [26]. For some $\delta \in (0, 1)$, we choose some differentiable monotone increasing penalty function $h \colon \mathbb{R} \to \mathbb{R}$ such that $h(x) > 0$ for $x > \delta$ and $h(x) = 0$ for $x \leqslant \delta$ and we add the term $\gamma h(|\mu_N|)$ for $\gamma > 0$ to the loss function that we introduce below, see Appendix A.2 for an example of $h$.

## 3.2 Adaptation with a generalised speed measure

Extending the objective from [54] to adapt the HMC proposal, we aim to solve

$$\arg \min_\theta \int \int \pi(q_0) \nu(v) \Big[ -\log a\left((q_0, v), (\mathcal{T}_L(v), \mathcal{W}_L(v))\right) + \beta \log r_L(\mathcal{T}_L(v)) \Big] \mathrm{d}v \mathrm{d}q_0, \quad (14)$$

where $\mathcal{T}_L$, $\mathcal{W}_L$, $r_L$ as well as the acceptance rate $a$ depend on $q_0$ and the parameters $\theta$ we want to adapt. Also, the hyper-parameter $\beta > 0$ can be adapted online by increasing $\beta$ if the acceptance rate is above a target acceptance rate $\alpha_\star$ and decreasing $\beta$ otherwise. We choose $\alpha_\star = 0.67$, which is optimal for increasing $d$ under independence assumptions [8]. One part of the objective constitutes minimizing the energy error $\Delta(q_0, v) = H(\mathcal{T}_L(v), \mathcal{W}_L(v)) - H(q_0, C^{-\top} v)$ that determines the log-acceptance rate via $\log a(q_0, C^{-\top} v) = \min\{0, -\Delta(q_0, v)\}$. Unbiased gradients of the energy error can be obtained without stopping any gradient calculations in the backward pass. However, we found that a multi-step extension of the biased fast MALA approximation from [54] tends to improve the adaptation by stopping gradients through $\nabla U$ as shown in Appendix A.3.

Suppose that the current state of the Markov chain is $q$. We resample the momentum $v \sim \mathcal{N}(0, \mathrm{I})$ and aim to solve (14) by taking gradients of the penalised loss function

$$-\min\{0, -\Delta(q, v)\} - \beta \left( d \log h + \log |\det C| + \mathcal{L}(\theta) - \gamma h(|\mu_N|) \right),$$

as illustrated in Algorithm 1, which also shows how we update the hyperparameters $\beta$ and $\gamma$. The adaptation scheme in Algorithm 1 requires to choose learning rates $\rho_\theta$, $\rho_\beta$, $\rho_\gamma$ and can be viewed within a stochastic approximation framework of controlled Markov chains, see for instance [2, 1, 3]. Different conditions have been established so that infinite adaptive schemes still converge to the correct invariant distribution, such as diminishing adaptation and containment [50]. We have used Adam [37] with a constant step size to adapt the mass matrix, but have stopped the adaptation after some fixed steps so that any convergence is preserved and we leave an investigation of convergence properties of an infinite adaptive scheme for future work.

## 4 Numerical experiments

This section illustrates the mixing performance of the entropy-based sampler for a variety of target densities. First, we consider Gaussian targets either in high dimensions or with a high condition number. Our results confirm (i) that HMC scales better than MALA for high-dimensional Gaussian targets and (ii) that the adaptation scheme learns a mass matrix that is adjusted to the geometry of the target. This is in contrast to adaptation schemes trying to optimize the ESJD [49] or variants thereof [40] that can lead to good mixing in a few components only. Next, we apply the novel adaptation scheme to Bayesian logistic regression models and find that it often outperforms NUTS, except in a few data sets where some components might mix less efficiently. We also compare the entropy-based adaptation with Riemann-Manifold based samplers for a Log-Gaussian Cox point process models. We find that both schemes mix similarly, which indicates that the gradient-based adaptation scheme can learn a suitable mass matrix without having access to the expected Fisher information matrix. Then, we consider a high-dimensional stochastic volatility model where the entropy-based scheme performs favourably compared to alternatives and illustrate that efficient sparsity assumptions can be accommodated when learning the mass matrix. Finally, we show in a toy example how the suggested

---

**Algorithm 1** Sample the next state $q'$ and adapt $\beta$, $\gamma$ and $\theta$.

---

1:  Sample velocity $v \sim \mathcal{N}(0, \mathrm{I})$ and set $p = C^{-\top} v$.
2:  Apply integrator LF to obtain $(q_\ell, p_\ell, \nabla U(q_\ell))_{0 \leqslant \ell \leqslant L} = \mathtt{LF}(q, p)$.
3:  Stop gradients $\nabla U(q_\ell) = \mathtt{stop\_grad}(\nabla U(q_\ell))$ for $0 \leqslant \ell \leqslant L$.
4:  Compute $\Xi_L(v)$ using (5).
5:  Compute $\Delta(q_0, v)$ using (16) and set $a = \min\{1, \mathrm{e}^{-\Delta(q_0, v)}\}$.
6:  Compute $\bar{\eta}_N, y = \textsc{Rademacher}()$.
7:  Set $\mathcal{L}(\theta) = \mathtt{stop\_grad}(y)^\top D_L \varepsilon$.
8:  Set $b_N = \mathtt{stop\_grad}\left(\frac{\bar{\eta}_N}{\|\bar{\eta}_N\|_2^2}\right)$ and $\mu_N = b_N^\top D_L b_N$.
9:  $\mathcal{E}(\theta) = -\min\{0, -\Delta(q_0, v)\} - \beta\left(d \log h + \log|\det C| + \mathcal{L}(\theta) - \gamma h(|\mu_N|)\right).$
10: Adapt $\theta \leftarrow \theta - \rho_\theta \nabla_\theta \mathcal{E}(\theta)$.
11: Adapt $\beta \leftarrow \Pi_\beta\left[\beta(1 + \rho_\beta(a - \alpha_\star)\right]$. #$\Pi_\beta$ projects onto a compact set; default value $[10^{-2}, 10^2]$.
12: Adapt $\gamma \leftarrow \Pi_\gamma\left[\gamma + \rho_\gamma h(|\mu_N|)\right]$. #$\Pi_\gamma$ projects onto a compact set; default value $[10^3, 10^5]$.
13: Sample $u \sim \mathcal{U}(0, 1)$ and set $q' = \mathbf{1}_{\{u \leqslant a\}} q_L + \mathbf{1}_{\{u > a\}} q$.

14: **function** $D_L(w)$:
15:     #$D_L(w) = D_L w$ computes Hessian-vector products efficiently
16:     $z = \mathtt{vjp}(\nabla U, \mathtt{stop\_grad}(q_{\lfloor L/2 \rfloor}), Cw)^\top$
17:     **return** $-h^2 \frac{L^2 - 1}{6} C^\top z$
18: **end function**

19: **function** $\textsc{Rademacher}$:
20:     Sample Rademacher random variable $\varepsilon$ and truncation level $N$.
21:     Initialise $y \leftarrow 0$ and $\bar{\eta}_0 = \varepsilon$.
22:     **for** $k = 1 \ldots N$ **do**
23:         #Apply a spectral normalisation for stability if $D_L$ is not a contraction; $\delta' \in (0, 1)$.
24:         Set $\bar{\eta}_k = D_L \bar{\eta}_{k-1} \cdot \min\left\{1, \delta' \|\bar{\eta}_{k-1}\|_2 / \|D_L \bar{\eta}_{k-1}\|_2\right\}$ and $y \leftarrow y + \frac{(-1)^k}{p_k} \bar{\eta}_k$.
25:     **end for**
26:     **return** $\bar{\eta}_N, y$
27: **end function**

---

approach might be modified to sample from highly non-convex potentials. Our implementation[2] builds up on tensorflow probability [39] with some target densities taken from [53]. We used 10 parallel chains throughout our experiments to adapt the mass matrix.

## 4.1 Gaussian targets

**Anisotropic Gaussian distributions.** We consider sampling from a multivariate Gaussian distribution $\mathcal{N}(0, \Sigma)$ with strictly convex potential $U(q) = \frac{1}{2} q^\top \Sigma^{-1} q$ for different covariance matrices $\Sigma$. For $c > 0$, assume a covariance matrix given by $\Sigma_{ij} = \delta_{ij} \exp\left(c(i - 1)/(d - 1) \log 10\right)$. We set (i) $c = 3$ and $d \in \{10^3, 10^4\}$ and (ii) $c = 6$ and $d = 100$, as considered in [52]. The eigenvalues of the covariance matrix are thus distributed between 1 to 100 in setting (i), while they vary from 1 and $10^6$ in setting (ii). The preconditioning factor $C$ is assumed to be diagonal. We adapt the sampler for $4 \times 10^4$ steps in case (i) and for $10^5$ steps in case (ii). We compared it with a NUTS implementation in tensorflow probability (TFP) [39] with a default maximum tree depth of 10 and step sizes adapted using dual averaging [32, 47] that we denote by N in the figures below. Additionally, we consider a further adaptation of NUTS by adapting a diagonal mass matrix using an online variance estimate of the accepted samples as implemented in TFP and denoted AN subsequently. We also consider two objectives as a replacement of the generalised speed measure (GSM): (a) the ESJD and (b) a weighted combination of the ESJD and its inverse as suggested in Levy et al. [40], without any burn-in component, which we denote L2HMC, see Appendix D for a precise definition. We compute the minimum and mean effective sample size (minESS and meanESS) of all functions $q \mapsto q_i$ over $i \in \{1, \ldots, d\}$ as shown in Figure 1a-1b for $d = 10^3$ in case (i) with leapfrog steps ranging from $L = 1$ to 10. It can be observed that HMC adapted with the GSM objective performs

---

[2]https://github.com/marcelah/entropy_adaptive_hmc

---

well in terms of minESS/sec for $L > 1$, whereas the ESJD or L2HMC objectives yield poor mixing as measured in terms of the minESS/sec. The meanESS/sec statistics are more similar for the different objectives. These observations provide some empirical evidence that the ESJD can be high even when some components mix poorly, which has been a major motivation for the GSM objective in [54]. The mass matrix learned using the GSM adapts to the target covariance as can be seen from the the condition numbers of $C^\top \Sigma^{-1} C$ in Figure 1c becoming relatively close to 1. The GSM objective also yields acceptance rates approaching 1 for increasing leap-frog steps and multiplication with $D_L$ becomes a contraction as shown in Appendix F.1, Figure 7. Results for $d = 10^4$ can be found in Figure 8 in Appendix F.1 which indicate that as the dimension increases, using more leap-frog steps becomes more advantageous. For the case (ii) of a very ill-conditioned target, results in Table 1 show that the GSM objective leads to better minESS/sec values, while further statistics shown in Figure 9 illustrate that the GSM also yields to higher minESS/sec values compared to NUTS with an adapted mass matrix. We want to emphasize that for fixed $L$, high acceptance rates for HMC need not be disadvantageous. This is illustrated in Figure 11 in Appendix F.4 for a Gaussian target $\mathcal{N}(0, I)$ in dimension $d = 10$, where tuning just the step-size to achieve a target acceptance rate can lead to slow mixing for some $L$, because the proposal can make a U-turn.

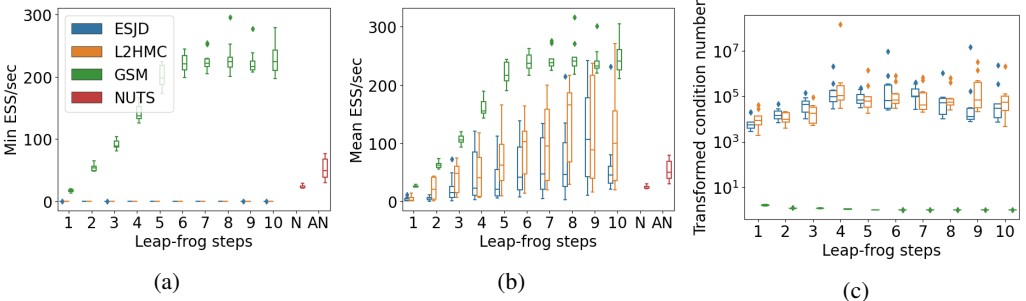

$$\text{(a)} \qquad\qquad\qquad \text{(b)} \qquad\qquad\qquad \text{(c)}$$

Figure 1: Minimum (1a) and mean (1b) effective sample size of $q \mapsto q_i$ per second after adaptation for an anisotropic Gaussian target ($d = 1000$). The condition number of the transformed Hessian $C^\top \Sigma^{-1} C$ are shown in (1c).

**Correlated Gaussian distribution.** We sample from a 51-dimensional Gaussian target with covariance matrix given by the squared exponential kernel plus small white noise as in [54], with $k(x_i, x_j) = \exp\left(-\frac{1}{2}(x_i - x_j)^2/0.4^2\right) + .01\delta_{ij}$ on the regular grid $[0, 4]$. We consider a general Cholesky factor $C$. The adaptation is performed over $10^5$ steps. Results over 10 runs are shown in Figure 10 in Appendix F.3 and summarized in Table 2.

Table 1: MinESS/sec for gradient-based adaptation schemes targeting an ill-conditioned Gaussian density ($d = 100$).

| Steps | GSM | ESJD | L2HMC |
|---|---|---|---|
| 1 | 122.3 (15.5) | 0.1 (0.01) | 0.1 (0.01) |
| 5 | 753.8 (22.2) | 0.1 (0.02) | 0.1 (0.02) |
| 10 | 570.0 (37.4) | 0.6 (395.2) | 0.1 (0.05) |

Table 2: MinESS/sec for gradient-based adaptation schemes targeting a correlated Gaussian density ($d = 51$).

| Steps | GSM | ESJD | L2HMC |
|---|---|---|---|
| 1 | 63.8 (3.9) | 0.8 (1.6) | 0.3 (0.1) |
| 5 | 390.0 (5.0) | 2.0 (5.4) | 2.7 (2.3) |
| 10 | 282.7 (7.8) | 0.9 (3.7) | 0.4 (0.9) |

## 4.2 Logistic regression

Consider a Bayesian logistic regression model with $n$ data points $y_i \in \{0, 1\}$ and $d$-dimensional covariates $x_i \in \mathbb{R}^d$ for $i \in \{1, \ldots, n\}$. Assuming a Gaussian prior with covariance matrix $\Sigma_0$ implies a potential function $U(q) = \sum_{i=1}^n \left[ -y_i x_i^\top q + \log\left(1 + e^{x_i^\top q}\right)\right] + \frac{1}{2} q^\top \Sigma_0^{-1} q$. We considered six datasets (Australian Credit, Heart, Pima Indian, Ripley, German Credit and Caravan) that are commonly used for benchmarking inference methods, cf. [16]. The state dimension ranges from $d = 3$ to $d = 87$. We choose $\Sigma_0 = I$ and parameterize $C$ via a Cholesky matrix. We adapt over $10^4$ steps. HMC with a moderate number of leap-frog steps tends to perform better for four out of

six data sets, with subpar performance for the Australian and Caravan data in terms of minESS/sec, albeit with higher mean ESS/sec across dimensions. The adaptive HMC algorithm tends to perform well if $D_L$ is contractive during iterations of the Markov chain such as for the German Credit data set as shown in Figure 2, where the eigenvalues of $D_L$ are estimated using a power iteration. If this is not the case as for the Caravan data in Figure 3, the adapted HMC algorithm can perform worse than MALA or NUTS. More detailed diagnostics for all data sets can be found in Appendix G.

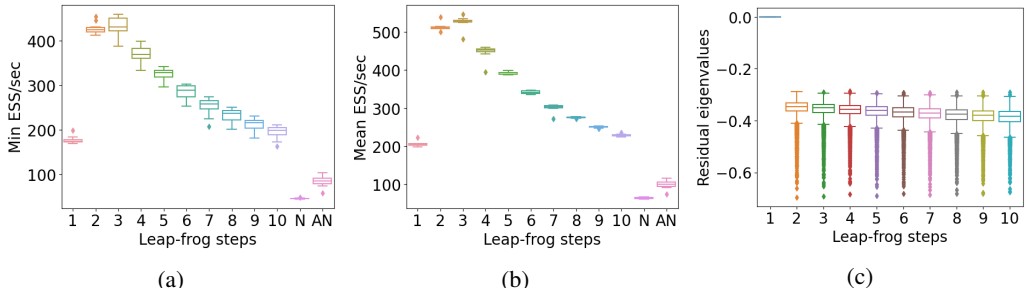

Figure 2: Minimum (2a) and mean (2b) effective sample size for a Bayesian logistic regression model for German credit data set ($d = 25$) after adaptation. Estimates of the eigenvalues of $D_L$ in 2c.

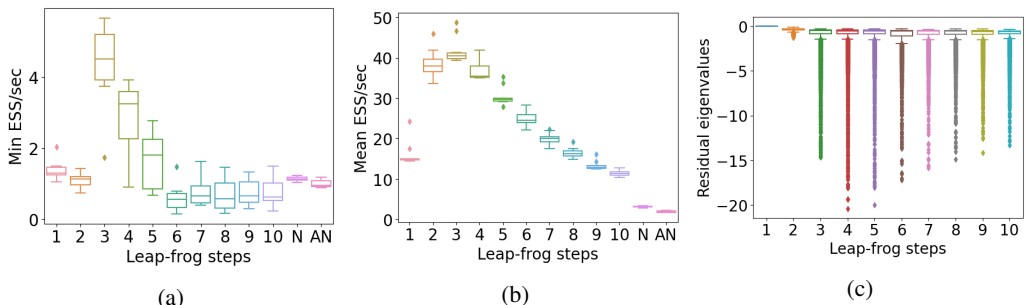

Figure 3: Minimum (3a) and mean (3b) effective sample size for a Bayesian logistic regression model for caravan data set ($d = 87$) after adaptation. Estimates of the eigenvalues of $D_L$ in 3c.

### 4.3 Log-Gaussian Cox Point Process

Inference in a log-Gaussian Cox process model is an ideal setting for Riemann-Manifold (RM) MALA and HMC [25], as a constant metric tensor is used therein that does not depend on the position, making the complexity no longer cubic but only quadratic in the dimension $d$ of the target. Consider an area on $[0, 1]^2$ discretized into grid locations $(i, j)$, for $i, j = 1, \ldots, n$. The observations $y_{ij}$ are Poisson distributed and conditionally independent given a latent intensity process $\{\lambda\}_{ij}$ with means $\lambda_{ij} = m \exp(x_{ij})$ for $m = n^{-2}$ and a latent vector $x$ drawn from a Gaussian process with constant mean $\mu$ and covariance function $\Sigma_{(i,j),(i',j')} = \sigma_x^2 \exp\{-\sqrt{(i - i')^2 + (j - j')^2}/(n\beta)\}$. The target is proportional to $p(y, x) \propto \prod_{i,j}^{n \times n} \exp\left[y_{ij}x_{ij} - m \exp(x_{ij})\right] \exp\left[-(x - \mu\mathbf{1})^\top \Sigma^{-1} (x - \mu\mathbf{1})/2\right]$. For the RM based samplers, the preconditioning matrix is $M = \Lambda + \Sigma^{-1}$ where $\Lambda$ is a diagonal matrix with diagonal elements $\{m \exp(\mu + \Sigma_{ii})\}_i$ and step sizes adapted using dual averaging. We generate simulated data for $d \in \{64, 256\}$ and adapt for 2000 steps using a Cholesky factor $C$. Figure 18 in Appendix H illustrates that the entropy-based adaptation can achieve a higher minESS/sec score for $d = 64$ with high acceptance rates for increasing leap-frog steps. The RM samplers perform slightly better in terms of minESS/sec for $d = 256$, see Figure 4 and Figure 19 for a comparison of the inverse mass matrices.

### 4.4 Stochastic volatility model

We consider a stochastic volatility model [36, 34] that has been used with minor variations for adapting HMC [25, 32, 57]. Assume that the latent log-volatilities follow an autoregressive AR(1)

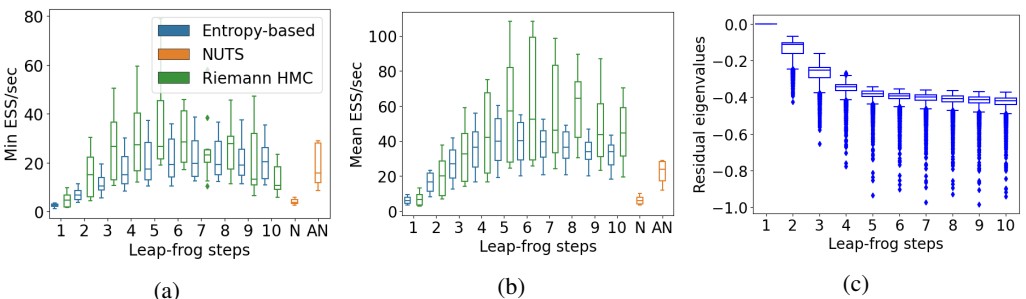

(a)          (b)          (c)

Figure 4: Minimum (4a) and mean (4b) effective sample size for a Cox process in dimension $d = 256$ after adaptation. Estimates of the eigenvalues of $D_L$ using power iteration in (4c).

process so that $h_1 \sim \mathcal{N}(0, \sigma^2/(1 - \phi^2))$ and for $t \in \{1, \dots, T-1\}$, $h_{t+1} = \phi h_t + \eta_{t+1}$ with $\eta_t \sim \mathcal{N}(0, \sigma^2)$. The observations follow the dynamics $y_t | h_t \sim \mathcal{N}(0, \exp(\mu + h_t))$. The prior distributions for the static parameters are: the persistence of the log-volatility process $(\phi + 1)/2 \sim \mathrm{Beta}(20, 1.5)$; the mean log-volatility $\mu \sim \mathrm{Cauchy}(0, 2)$; and the scale of the white-noise process $\sigma \sim \mathrm{Half\text{-}Cauchy}(0, 1)$. We reparametrize $\phi$ and $\sigma$ with a sigmoid- and softplus-transformation, respectively. Observe that the precision matrix of the AR(1) process is tridiagonal. Since a Cholesky factor of such a matrix is tridiagonal, we consider the parameterization $C = B_\theta^{-1}$ for an upper-triangular and tridiagonal matrix $B_\theta$. The required operations with such banded matrices have a complexity of $\mathcal{O}(d)$, see for instance [22]. For comparison, we also consider a diagonal matrix $C$. We apply the model to ten years of daily returns of the S&P500 index, giving rise to a target dimension of $d = 2519$. In order to account for the different number of gradient evaluations, we use $3.5 \times 10^4/L$ steps for the adaptation and for evaluating the sampler based on $L \in \{1, \dots, 10\}$ leapfrog steps. We run NUTS for 1000 steps which has a four times higher run-time compared to the other samplers. In addition to using effective sample size to assess convergence, we also report the potential scale reduction factor split-$\hat{R}$ [24, 55] where large values are indicative of poor mixing. We report results over three replications in Figure 5 with more details in Figure 20, Appendix I. First, HMC with moderately large $L$ tends to improve the effective samples per computation time compared to the MALA case, while also having a smaller $\hat{R}$. Second, using a tridiagonal mass matrix improves mixing compared to a diagonal one, particularly for the latent log-volatilities as seen in the median ESS/sec or median $\hat{R}$ values. The largest absolute eigenvalue of $D_L$ tends to be smaller for a tridiagonal mass matrix and the acceptance rates are approaching $100\%$ more slowly for increasing $L$. Third, NUTS seems less efficient as does using a dual-adaptation scheme.

We imagine that similar efficient parameterizations of $M$ or $M^{-1}$ can be used for different generalisations of the above stochastic volatility model, such as including $p$ sub-diagonals for log-volatilities having a higher-order AR($p$) dynamics or multivariate extensions using a suitable block structure. Likewise, this approach might also be useful for inferences in different Gaussian Markov Random Field models with sparse precision matrices.

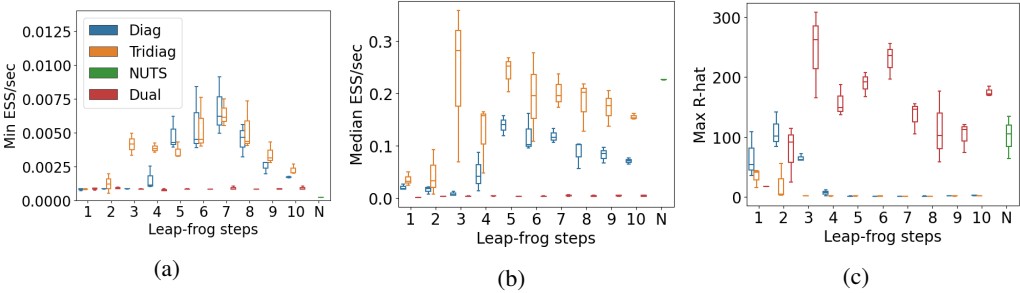

(a)          (b)          (c)

Figure 5: Minimum (5a) and median (5b) effective sample size per second and maximum $\hat{R}$ of $q \mapsto q_i$ for a stochastic volatility model ($d = 2519$) after adaptation.

### 4.5 Learning non-linear transformations

To illustrate an extension to sample from highly non-convex targets by learning a non-linear transformation within the suggested framework as explained in greater detail in Appendix C, we consider sampling from a two-dimensional Banana distribution that results from the transformation of $\mathcal{N}(0, \Lambda)$ where $\Lambda$ is a diagonal matrix having entries $100$ and $1$ via the volume-preserving map $\phi_b(x) = (x_1, x_2 + b(x_1^2 - 100))$, for $b = 0.1$, cf. [27]. We consider a RealNVP-type [19] transformation $f = f_3 \circ f_2 \circ f_1$ where $f_1(x_1, x_2) = (x_1, x_2 \cdot g(s(x_1)) + t(x_1))$, $f_2(x_1, x_2) = (x_1 \cdot g(s(x_2)) + t(x_1), x_2)$ and $f_3(x_1, x_2) = (c_1 x_1, c_2 x_2)$. The functions $s$ and $t$ are neural networks with two hidden layers of size 50. For numerical stability, we found it beneficial to use a modified affine scaling function $g$ as a sigmoid function scaled on a restricted range such as $(0.5, 2)$, as also suggested in [6]. As an alternative, we also consider learning a linear transformation $f(x) = Cx$ for a Cholesky matrix $C$ as well as NUTS and a standard HMC sampler with step size adapted to achieve a target acceptance rate of $0.65$. Figure 6 summarizes the ESS where each method uses $4 \times 10^5$ samples before and after the adaptation. Whereas a linear transformation does not improve on standard HMC, non-linear transformations can improve the mixing efficiency.

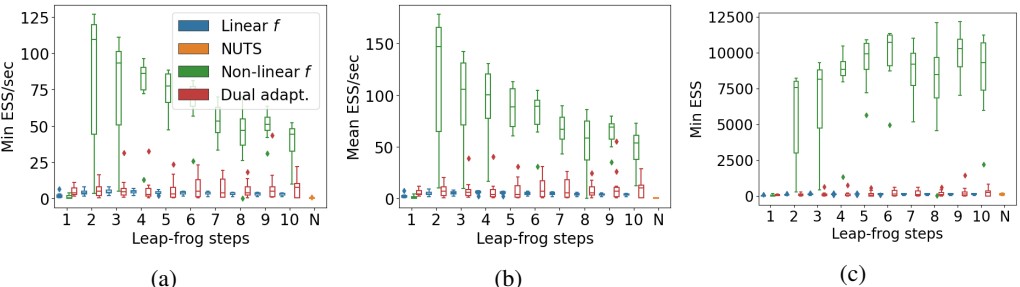

Figure 6: Minimum (6a) and mean (6b) effective sample size per second as well as minimum effective sample size (6c) for a Banana-shaped target in dimension $d = 2$ after adaptation.

## 5 Discussion and Outlook

**Limitations.** Our approach to learn a constant mass matrix can struggle for targets where the Hessian varies greatly across the state space, which can yield relatively short integration times with very high acceptance rates. While this effect might be mitigated by considering non-linear transformations, it remains challenging to learn flexible transformations efficiently in high dimensions.

**Variations of the entropy objective.** Recent work [18, 11] have suggested to add the cross-entropy term $\int \pi(q) \int r(q'|q) \log \pi(q') \mathrm{d}q' \mathrm{d}q$ to the entropy objective for optimizing the parameters of a Metropolis-Hastings kernel with proposal density $r(q'|q)$. Algorithm 1 can be adjusted to such variations, possibly by stopping gradients through $\nabla U$ as for optimizing the energy error term.

**Variations of HMC.** We have considered a standard HMC setting for a fixed number of leap-frog steps. One could consider a mixture of HMC kernels with different numbers of leap-frog steps and an interesting question would be how to learn the different mass matrices jointly in an efficient way.

Instead of a full velocity refreshment, partial refreshment strategies [33] can sometimes mix better. The suggested adaptation approach can yield very high acceptance rates particularly for increasing leap-frog steps and the learned mass matrix can be used with a partial refreshment. However, it would be interesting to analyse if the adaptation can be adjusted to such persistent velocity updates. It would also be of interest to analyse if similar ideas can be used to adapt different numerical integrators such as those suggested in [7] for target densities relative to a Gaussian measure or for multinomial HMC with an additional intra-trajectory sampling step [9, 59].

Our focus was on learning a mass matrix so that samples from the Markov chain can be used for estimators that are consistent for increasing iterations. However, unbiased estimators might also be constructed using coupled HMC chains [30] and one might ask if the adapted mass matrix leads to shorter meeting times in such a setting.

## Acknowledgements

The authors acknowledge the use of the UCL Myriad High Performance Computing Facility (Myriad@UCL), and associated support services, in the completion of this work.

## Funding Transparency Statement

There are no additional sources of funding to disclose.

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
