# Appendices

## A  Gradient terms for the adaptation scheme

### A.1  Gradients for the entropy approximation

Following the arguments in [13], we can compute the gradient of the term in (13) using

$$\frac{\partial}{\partial \theta_i} \mathcal{L}(\theta) = \mathrm{Tr}\left(\sum_{k=0}^{\infty}(-1)^k [D_L]^k \frac{\partial}{\partial \theta_i}\{D_L\}\right) = \mathbb{E}_{N,\varepsilon}\left[\sum_{k=0}^{N}\frac{(-1)^k}{p_k}\varepsilon^{\top}[D_L]^k \frac{\partial}{\partial \theta_i}\{D_L\}\varepsilon\right],$$

which yields a stochastic gradient via a Russian-roulette estimator.

Additionally, to avoid gradients with infinite means even if $D_L$ is not contractive, we consider a spectral normalisation, so that instead of computing recursively $\eta_0 = \varepsilon$ and $\eta_k = D_L\eta_{k-1}$ for $k \in \{1, \ldots, N\}$, we set $\bar{\eta}_0 = \varepsilon$ and

$$\bar{\eta}_k = D_L\bar{\eta}_{k-1} \cdot \min\{1, \delta' \|\bar{\eta}_{k-1}\|_2 / \|D_L\bar{\eta}_{k-1}\|_2\} \tag{15}$$

for $k \in \{1, \ldots, N\}$ and $\delta' \in (0,1)$, such as $\delta' = 0.99$ in all our experiments. We obtain an estimator

$$\frac{\partial}{\partial \theta_i} \mathcal{L}(\theta) \approx \mathbb{E}_{N,\varepsilon}\left[\sum_{k=0}^{N}\frac{(-1)^k}{p_k}\bar{\eta}_k^{\top} \frac{\partial}{\partial \theta_i}\{D_L\}\varepsilon\right].$$

### A.2  Gradients for the penalty function

We used the following penalty function

$$h(x) = (x-\delta)^2 \mathbb{1}_{\{x\in[\delta,\delta_2)\}} + ((\delta_2-\delta)^2 + (\delta_2-\delta)^2(x-\delta_2))\mathbb{1}_{\{x\geqslant\delta_2\}}$$

throughout our experiments with $\delta \in \{0.75, 0.95\}$, and $\delta_2 = 1 + \delta$. The motivation was to have a quadratic increase for the penalty term if the largest absolute eigenvalue approaches 1, and then smoothly switch to a linear function for values larger than $\delta_2$. Gradients for this function can be computed routinely using automatic differentiation.

### A.3  Gradients for the energy error

We can write the energy error as

$$\Delta(q_0, v) = U(\mathcal{T}_L(v)) - U(q_0) + K(\mathcal{W}_L(v)) - K(C^{-\top}v)$$

$$= U\left(q_0 + LhCv - h^2 CC^{\top}\Xi_L(v) - L\frac{h^2}{2}CC^{\top}\nabla U(q_0)\right) - U(q_0)$$

$$+ \frac{1}{2}\left\|v - \frac{h}{2}C[\nabla U(q_0) + \nabla U(q_L)] - hC\sum_{\ell=1}^{L-1}\nabla U(q_\ell)\right\|^2 - \frac{1}{2}\|v\|^2. \tag{16}$$

Recall from (5) that $\Xi_L(v)$ is a weighted sum of potential energy gradients along the leap-frog trajectory. For computing gradients of the energy-error for the fast approximation, we therefore stop the gradient for all $\nabla U(q_\ell)$ for any $\ell \in \{1, \ldots, L\}$.

## B  Proof of Lemma 1

*Proof.* We generalise the arguments from [14], Lemma 7. Proceeding by induction over $n$, we have for the case $n = 1$, for any $v \in \mathbb{R}^d$, that $\mathsf{D}\mathcal{T}_1(v) = hC$ and $\mathcal{S}_1(v) = \frac{1}{h}C^{-1}q_0 - \frac{h}{2}C^{\top}\nabla U(q_0)$ with derivative of zero. For the case $n = 2$, using (3) and (5), one obtains

$$\mathsf{D}\mathcal{T}_2(v) - 2hC - h^3 CC^{\top}\nabla^2 U(\mathcal{T}_1(v))C \tag{17}$$

and moreover
$$\mathsf{D}\mathcal{S}_2(v) = -\frac{h^2}{2}C^\top \nabla^2 U(\mathcal{T}_1(v))C \tag{18}$$
which establishes (10). Clearly, $\|\mathsf{D}\mathcal{S}_2(v)\|_2 < \frac{1}{8}$ if $2^2 h^2 < \frac{1}{4\|C^\top \nabla^2 U(\mathcal{T}_1(v))C\|_2}$.
Further, for any $n < L$, again from (3) and (5),

$$\begin{aligned}
\mathsf{D}\mathcal{T}_{n+1}(v) &= (n+1)hC - h^2 CC^\top \mathsf{D}\Xi_{n+1}(v) \\
&= (n+1)hC - h^2 CC^\top \left[\sum_{i=1}^n (n+1-i)\nabla^2 U(\mathcal{T}_i(v))\mathsf{D}\mathcal{T}_i(v)\right] \\
&= (n+1)hC - h^2 CC^\top \left[\sum_{i=1}^n (n+1-i)\nabla^2 U(\mathcal{T}_i(v))ihC\left(\mathrm{I}+\mathsf{D}\mathcal{S}_i(v)\right)\right] \\
&= (n+1)hC + (n+1)hC\left[-h^2 \sum_{i=1}^n \frac{(n+1-i)}{n+1}iC^\top \nabla^2 U(\mathcal{T}_i(v))C\left(\mathrm{I}+\mathsf{D}\mathcal{S}_i(v)\right)\right],
\end{aligned}$$

which shows the representation (10) for the case $n+1$ by recalling that
$$\mathsf{D}\mathcal{T}_{n+1}(v) = (n+1)hC(\mathrm{I}+\mathsf{D}\mathcal{S}_{n+1}(v)).$$
Assume now that $\|\mathsf{D}\mathcal{S}_\ell(v)\|_2 < 1/8$ holds for all $\ell \leqslant n$. Then for any $v \in \mathbb{R}^d$

$$\begin{aligned}
\|\mathsf{D}\mathcal{S}_{n+1}(v)\|_2 &\leqslant \frac{h^2}{n+1}\sum_{i=1}^n i(n+1-i)\left\|C^\top \nabla^2 U(\mathcal{T}_i(v))C\right\|_2 \|\mathrm{I}+\mathsf{D}\mathcal{S}_i(v)\|_2 \\
&\leqslant \frac{h^2}{n+1}\sum_{i=1}^n \frac{L^2}{4}\left\|C^\top \nabla^2 U(\mathcal{T}_i(v))C\right\|_2 \|\mathrm{I}+\mathsf{D}\mathcal{S}_i(v)\|_2 \\
&\leqslant \frac{h^2}{n+1}\sum_{i=1}^n \frac{L^2}{4}\frac{1}{4L^2 h^2}\left(1+\frac{1}{8}\right) \leqslant \frac{1}{8}
\end{aligned}$$

where the second inequality follows from $(n+1-i)i \leqslant (\frac{n+1-i+i}{2})^2 \leqslant \frac{L^2}{4}$, whereas the third inequality follows from the induction hypothesis and the assumption $L^2 h^2 < \sup_q \frac{1}{4\|C^\top \nabla^2 U(q)C\|_2}$. $\square$

## C   Extension to learn non-linear transformations

The suggested approach can perform poorly for non-convex potentials or even convex potentials such as arsing in a logistic regression model for some data sets. We illustrate here how to learn a reasonable proposal for a general potential function by considering some version of position-dependent preconditioning. Consider an invertible differentiable transformation $f\colon \mathbb{R}^d \to \mathbb{R}^d$. The idea now is to run HMC with unit mass matrix for the transformed variables $z = f^{-1}(q)$ where $q \sim \pi$. Write $\tilde{\pi}$ for the density of $z$ and let $\tilde{U}$ be the corresponding potential energy function which is given by
$$\tilde{U}(z) = U(H(z)) - \log|\det \mathsf{D}f(z)|$$
with gradient
$$\nabla \tilde{U}(z) = \mathsf{D}f(z)^\top \nabla U(f(z)) - \nabla \log|\det \mathsf{D}f(z)|.$$
The transformation $f$ as well as $\tilde{U}$ generally depend on some parameters $\theta$ that we again omit for a less convoluted notation. Our approach can be seen as an alternative for instance to [31] where such a transformation is first learned by trying to approximate $\tilde{\pi}$ with a standard Gaussian density using variational inference, while the HMC hyperparameters are adapted in a second step using Bayesian optimisation.

We write $\tilde{\mathcal{T}}_L\colon v \mapsto z_L$ for the transformation that maps the initial velocity $v = p_0 \sim \mathcal{N}(0,\mathrm{I})$ to the $L$-th leapfrog step $z_L$, starting at $z_0$ based on the potential function $\tilde{U}$ with unit mass matrix $M = \mathrm{I}$. Analogously, we define the mapping $\tilde{\mathcal{W}}_L\colon v \mapsto p_L$ and similarly to (7), we define
$$\tilde{\mathcal{S}}_L(v) = \frac{1}{Lh}\tilde{\mathcal{T}}_L(v) - v.$$

We can then reparametrize the proposal at the point $q_0 = f(z_0)$ by $v \mapsto f(\tilde{\mathcal{T}}_L(v))$. Consequently, the log-density of the proposal is given by

$$\log r_L(f(\tilde{\mathcal{T}}_L(v))) = \log \nu(v) - \log |\det \mathsf{D}f(\tilde{\mathcal{T}}_L(v))| - \log |\det \mathsf{D}\tilde{\mathcal{T}}_L(v)|,$$

and we can write

$$\log |\det \mathsf{D}\tilde{\mathcal{T}}_L(v)| = d \log Lh + \log |\det(\mathrm{I} + \mathsf{D}\tilde{\mathcal{S}}_L(v))|.$$

We use the same approximation

$$\mathsf{D}\tilde{\mathcal{S}}_L(v) \approx -h^2 \frac{L^2 - 1}{6} \nabla^2 \tilde{U}(z_{\lfloor L/2 \rfloor})$$

based on the transformed Hessian now.

Hessian-vector products can similarly be computed using vector-Jacobian products: With $g(z) = \mathtt{grad}(\tilde{U}, z)$, we then compute $\nabla^2 \tilde{U}(z)w = \mathtt{vjp}(g, z, w)^\top$ for $z = f^{-1}(\mathtt{stop\_grad}(f(z_{\lfloor L/2 \rfloor})))$. The motivation for stopping the gradients comes from considering the special case $f \colon z \mapsto Cz$ that corresponds to the position-independent preconditioning scheme above. For such a linear transformation,

$$\tilde{U}(z) = C^\top \nabla^2 U(Cz) C.$$

To recover the previous case, we stop gradients at $q_{\lfloor L/2 \rfloor} = f(z_{\lfloor L/2 \rfloor}) = Cz_{\lfloor L/2 \rfloor}$.

Gradients for the log-accept ratio can be computed based on the log-accept ratio of the transformed chain [35]. The energy error of the transformed chain is

$$
\begin{aligned}
\Delta_\theta(q_0, v) =& U_\theta(\tilde{\mathcal{T}}_L(v)) - U_\theta(f^{-1}(q_0)) + K(\tilde{\mathcal{W}}_L(v)) - K(v) \\
=& U \Big\{ f \Big[ f^{-1}(q_0) + Lhv - h^2 \tilde{\Xi}_L(v) \\
& \qquad - L\frac{h^2}{2} \left( \mathsf{D}f(f^{-1}(q_0))^\top \nabla U(q_0) - \nabla \log |\det \mathsf{D}f(f^{-1}(q_0))| \right) \Big] \Big\} \\
& + \log |\det \mathsf{D}f(z_L)| - U(q) + \log |\det \mathsf{D}f(f^{-1}(q))| \\
& + \frac{1}{2} \Big\| v - \frac{h}{2} \big[ \mathsf{D}f(z_0)^\top \nabla U(f(z_0)) - \nabla \log |\det \mathsf{D}f(z_0) + \mathsf{D}f(z_L)^\top \nabla U(f(z_L)) \\
& \qquad - \nabla \log |\det \mathsf{D}f(z_L)| \big] - h \sum_{\ell=1}^{L-1} \mathsf{D}f(z_\ell)^\top \nabla U(f(z_\ell)) - \nabla \log |\det \mathsf{D}f(z_\ell)| \Big\|^2 \\
& - \frac{1}{2} \|v\|^2,
\end{aligned}
$$

where

$$\tilde{\Xi}_L(v) = \sum_{i=1}^{L} (L - i) \big[ \mathsf{D}f(z_i)^\top \nabla U(f(z_i)) - \nabla \log |\det \mathsf{D}f(z_i)| \big]$$

and $z_0, \ldots, z_L$ is the leap-frog trajectory starting at $z_0 = f^{-1}(q_0)$. We also stop all $U$ gradients, i.e. $\nabla U(f(z_\ell)) \leftarrow \mathtt{stop\_grad}(\nabla U(f(z_\ell)))$. It can be seen that this recovers the above setting if $f \colon z \mapsto Cz$.

## D   Gradient-based adaptation using the expected squared jumping distance and variations

We consider the different loss functions

$$\mathcal{F}_{\text{GSM}}(\theta) = -\int\int \pi(q_0)\nu(v)\Big[\log a\{(q_0,v),(\mathcal{T}_L(v),\mathcal{W}_L(v))\} - \beta\log r_L(\mathcal{T}_L(v))\Big]\mathrm{d}v\mathrm{d}q_0 \tag{19}$$

$$\mathcal{F}_{\text{ESJD}}(\theta) = -\int\int \pi(q_0)\nu(v)\Big[a\{(q_0,v),(\mathcal{T}_L(v),\mathcal{W}_L(v))\}\,\|q_0-\mathcal{T}_L(v)\|^2\Big]\mathrm{d}v\mathrm{d}q_0 \tag{20}$$

$$\mathcal{F}_{\text{L2HMC}}(\theta) = -\int\int \pi(q_0)\nu(v)\Big[\frac{a\{(q_0,v),(\mathcal{T}_L(v),\mathcal{W}_L(v))\}\,\|q_0-\mathcal{T}_L(v)\|^2}{\lambda} \tag{21}$$

$$-\frac{\lambda}{a\{(q_0,v),(\mathcal{T}_L(v),\mathcal{W}_L(v))\}\,\|q_0-\mathcal{T}_L(v)\|^2}\Big]\mathrm{d}v\mathrm{d}q_0.$$

The L2HMC objective (21) has been suggested in Levy et al. [40] for learning generalisations of HMC, although we ignore a burn-in term that has been included originally. In our implementation, we adapt $\lambda > 0$ online as a moving average of the expected squared jumping distance. The objectives (20) and (21) can be optimized using stochastic gradient descent similar to Algorithm 1 without the approximations as required for the GSM objective (19).

## E   Proof of the HMC proposal reparameterizations

For completeness, we provide a proof of the reparameterization (3) and (4) of the $L$-th step position $q_L$ and momentum $p_L$ using the velocity $v$ that relates to the initial momentum $p_0 \sim \mathcal{N}(0, M)$ via $p_0 = C^{-\top}v$. Such representations with an identity mass matrix have been used previously in [42, 21, 14].

*Proof.* We proceed by induction over $\ell \in \{1, \ldots, L\}$. For the case $\ell = 1$, the recursions in (2) imply

$$q_1 = q_0 + hCC^\top\Big[p_0 - \frac{h}{2}\nabla U(q_0)\Big] = q_0 + hCv - \frac{h}{2}CC^\top\nabla U(q_0)$$

and

$$p_1 = \Big[p_0 - \frac{h}{2}\nabla U(q_0)\Big] - \frac{h}{2}\nabla U(q_1) = C^{-\top}v - \frac{h}{2}\left[\nabla U(q_0) + \nabla U(q_1)\right].$$

Suppose now that the representations hold for $1 \leqslant \ell < L$. Then, using the recursions in (2),

$$q_{\ell+1} = q_\ell + hCC^\top\Big[p_\ell - \frac{h}{2}\nabla U(q_\ell)\Big]$$

$$= q_0 - \Big[\frac{\ell h^2}{2}CC^\top + \frac{h}{2}CC^\top\Big]\nabla U(q_0) + \left[\ell hC + hCC^\top C^{-\top}\right]v - h^2CC^\top\nabla U(q_\ell)$$

$$- h^2CC^\top\sum_{i=1}^{\ell-1}\nabla U(q_i) - h^2CC^\top\Xi_\ell(v)$$

$$= q_0 - \Big[(\ell+1)\frac{h^2}{2}CC^\top\Big]\nabla U(q_0) + (\ell+1)hCv - h^2CC^\top\sum_{i=1}^{\ell}\nabla(\ell+1-i)\nabla U(q_i).$$

This establishes the representation for $q_L$. The induction step for the momentum is a straightforward application of (2) to the induction hypothesis.

$\square$

# F  Gaussian targets experiments

## F.1  High-dimensional Gaussian targets

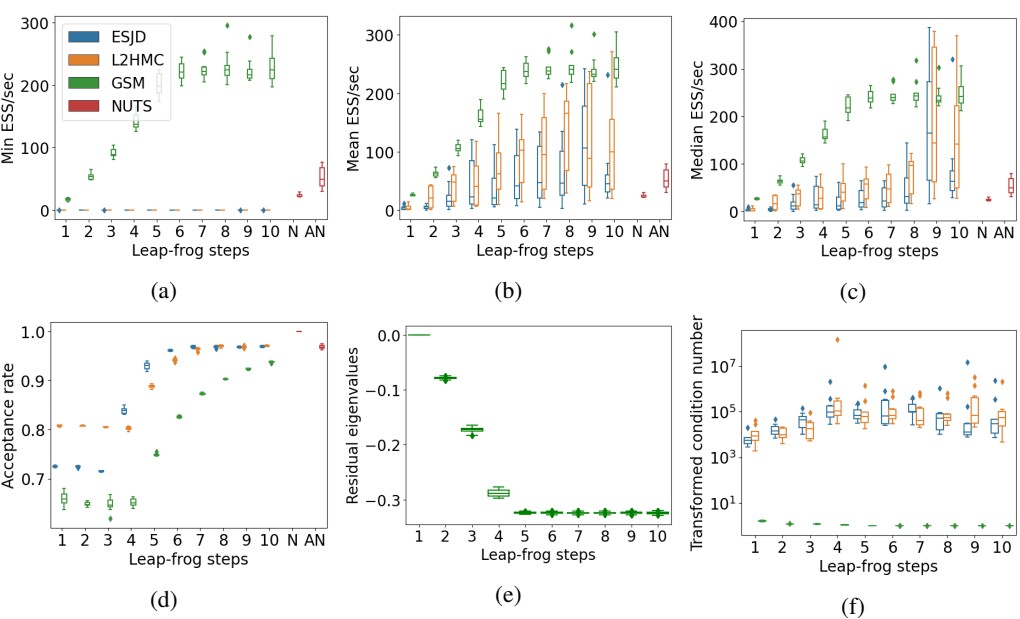

Figure 7: Anisotropic Gaussian target ($d = 1000$). Minimum (7a), mean (7b) and median (7c) effective sample size of $q \mapsto q_i$ per second. Average acceptance rates in 7d and estimates of the eigenvalues of $D_L$ in 7e. Condition number of transformed Hessian $C^\top \Sigma^{-1} C$ in 7f.

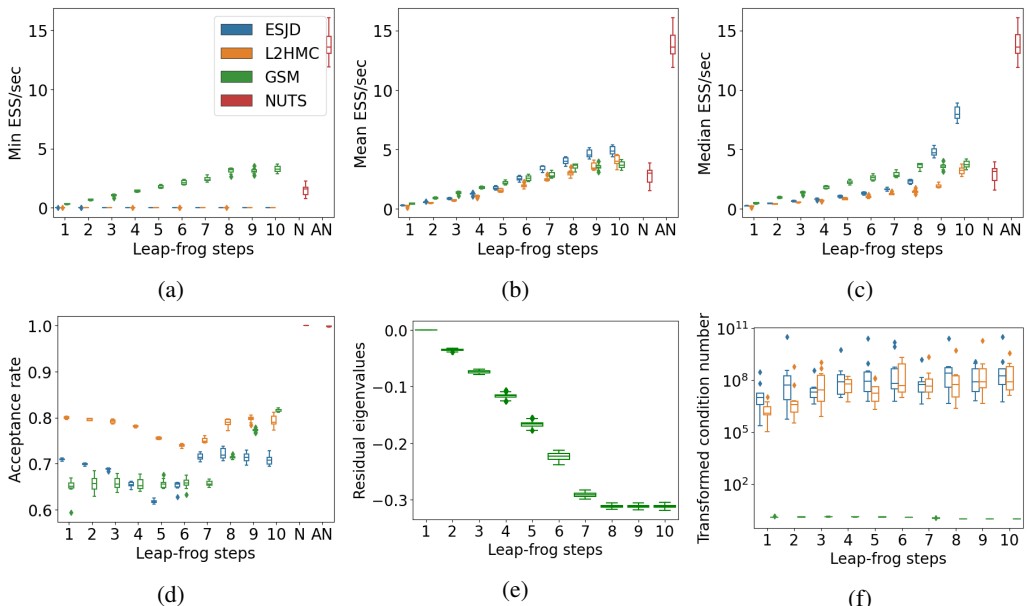

Figure 8: Independent Gaussian target ($d = 10000$). Minimum (8a), mean (8b) and median (8c) effective sample size of $q \mapsto q_i$ per second. Average acceptance rates in 8d and estimates of the eigenvalues of $D_L$ in 8e. Condition number of transformed Hessian $C^\top \Sigma^{-1} C$ in 8f.

## F.2 Ill-conditioned anisotropic Gaussian target

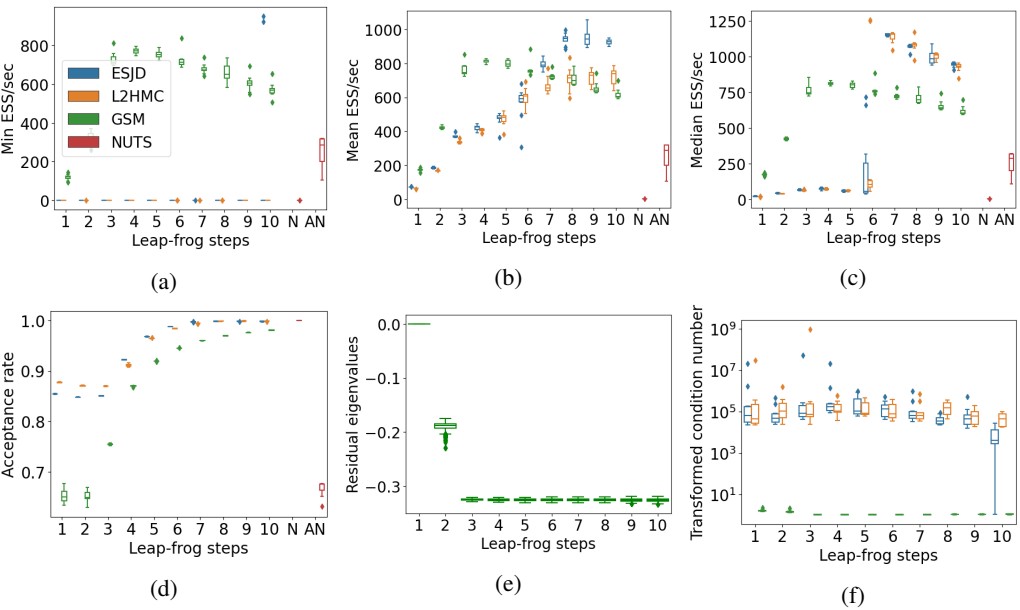

Figure 9: Ill-conditioned Gaussian target ($d = 100$). Minimum (9a), mean (9b) and median (9c) effective sample size of $q \mapsto q_i$ per second. Average acceptance rates in 9d and estimates of the eigenvalues of $D_L$ using power iteration in 9e. Condition number of transformed Hessian $C^\top \Sigma^{-1} C$ in 9f. Values computed after adaptation.

## F.3 Correlated Gaussian target

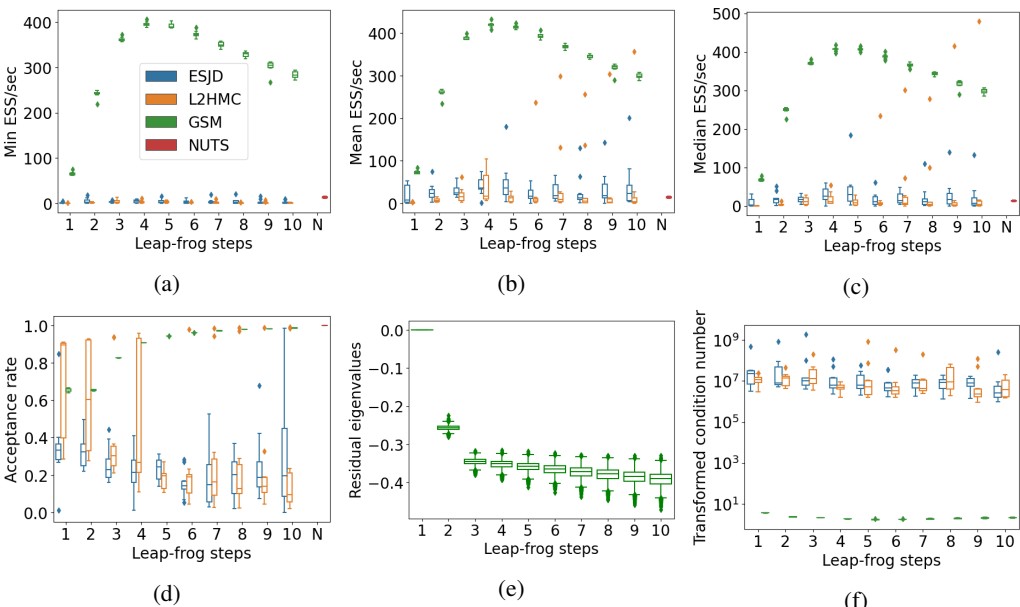

Figure 10: Correlated Gaussian target ($d = 51$). Minimum (10a), mean (10b) and median (10c) effective sample size of $q \mapsto q_i$ per second. Average acceptance rates in 10d and estimates of the eigenvalues of $D_L$ using power iteration in 10e. Condition number of transformed Hessian $C^\top \Sigma^{-1} C$ in 10f. Values computed after adaptation.

## F.4 IID Gaussian target

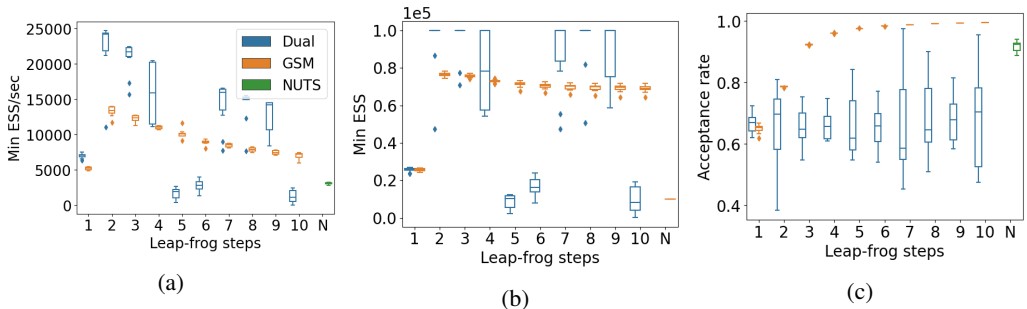

(a)      (b)      (c)

Figure 11: IID Gaussian target ($d = 10$). Minimum effective sample size of $q \mapsto q_i$ per second in 11a and absolute minimum effective sample size where NUTS is run for $1/10$-th of the iterations of the other schemes in 11b. Average acceptance rates in 11c. Values computed after adaptation.

# G Logistic regression experiments

## G.1 Australian credit data

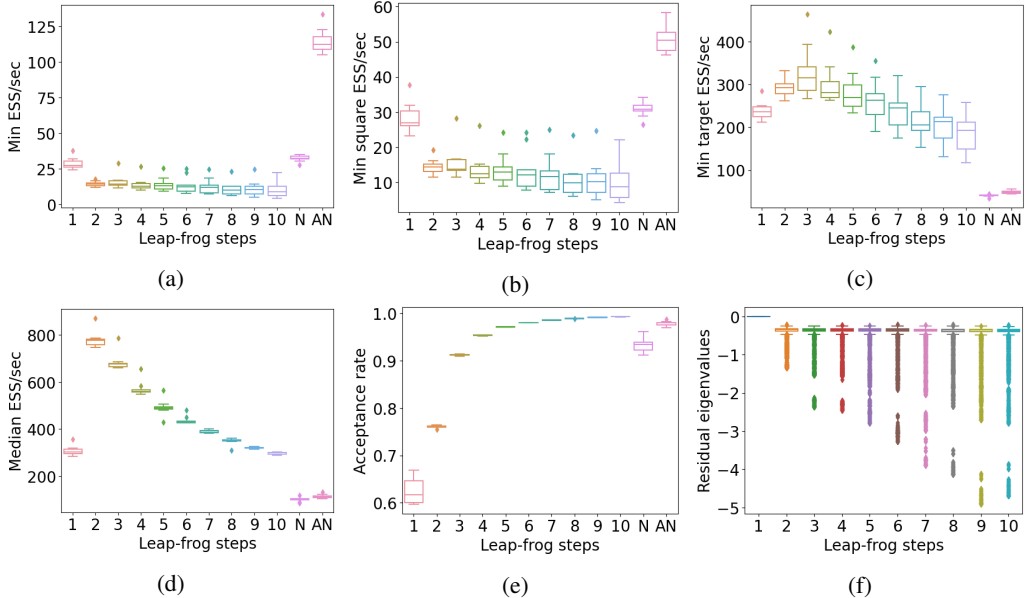

(a)      (b)      (c)

(d)      (e)      (f)

Figure 12: Bayesian logistic regression for Australian Credit data set ($d = 15$). Minimum effective sample size per second after adaptation of $q \mapsto q_i$ in 12a, of $q \mapsto q_i^2$ in 12b and of $q \mapsto \log \pi(q)$ in 12b. Median marginal effective sample per second in 12d and average acceptance rates in 12e and estimates of the eigenvalues of $D_L$ in 12f.

## G.2  Heart data

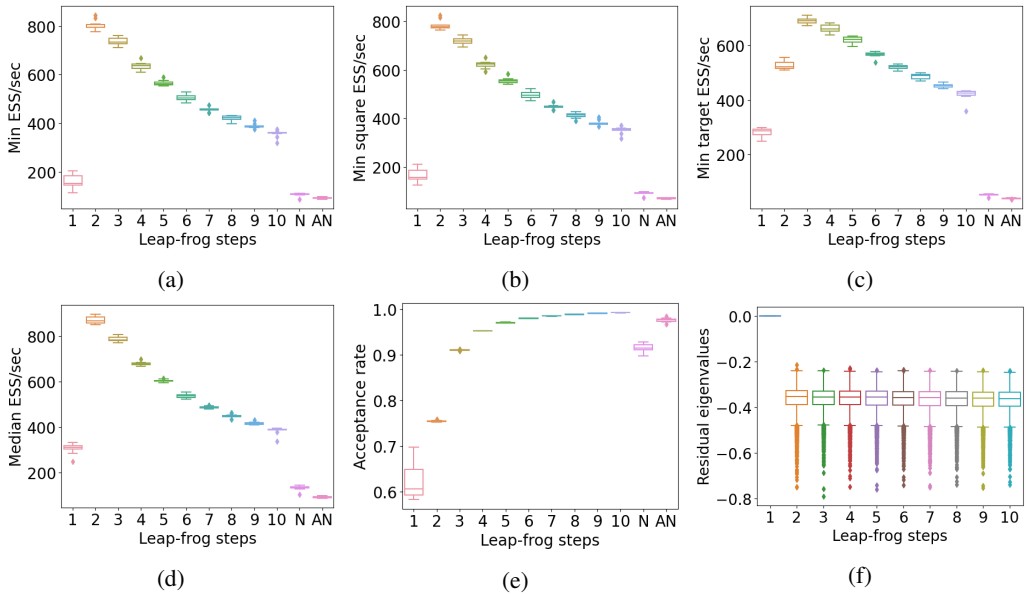

(a)  (b)  (c)

(d)  (e)  (f)

Figure 13: Bayesian logistic regression for heart data set ($d = 14$). Minimum effective sample size per second after adaptation of $q \mapsto q_i$ in 13a, of $q \mapsto q_i^2$ in 13b and of $q \mapsto \log \pi(q)$ in 13b. Median marginal effective sample per second in 13d and average acceptance rates in 13e and estimates of the eigenvalues of $D_L$ in 13f.

## G.3  Pima data

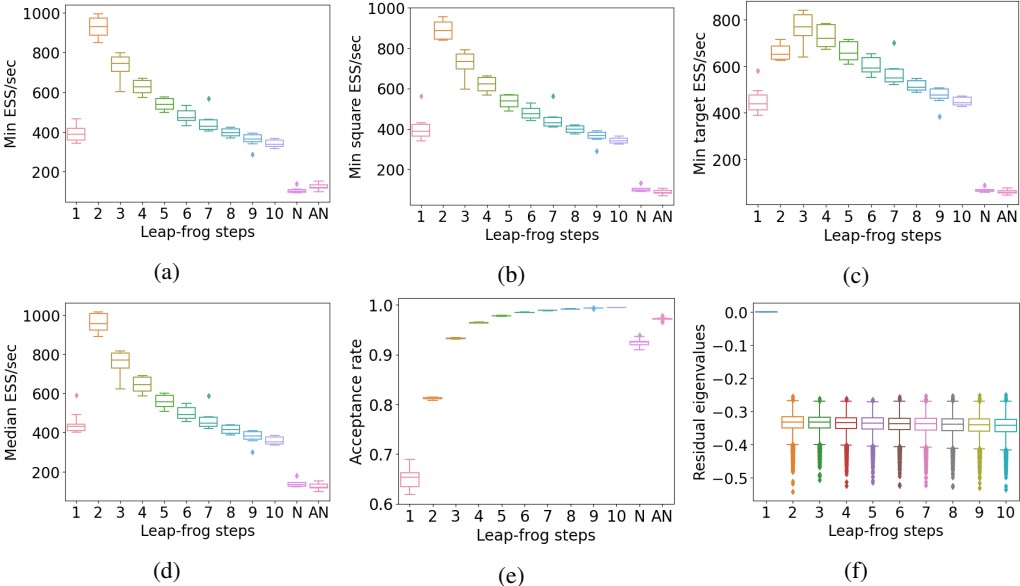

(a)  (b)  (c)

(d)  (e)  (f)

Figure 14: Bayesian logistic regression for Pima data set ($d = 8$). Minimum effective sample size per second after adaptation of $q \mapsto q_i$ in 14a, of $q \mapsto q_i^2$ in 14b and of $q \mapsto \log \pi(q)$ in 14c. Median marginal effective sample per second in 14d and average acceptance rates in 14e and estimates of the eigenvalues of $D_L$ in 14f.

## G.4  Ripley data

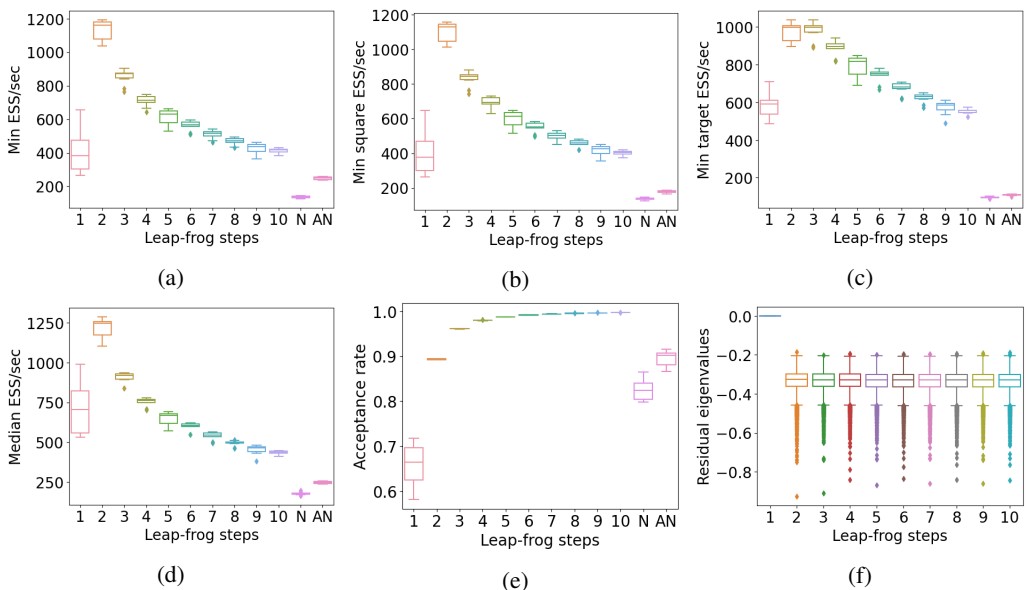

(a)  (b)  (c)

(d)  (e)  (f)

Figure 15: Bayesian logistic regression for Ripley data set ($d = 3$). Minimum effective sample size per second after adaptation of $q \mapsto q_i$ in 15a, of $q \mapsto q_i^2$ in 15b and of $q \mapsto \log \pi(q)$ in 15c. Median marginal effective sample per second in 15d and average acceptance rates in 15e and estimates of the eigenvalues of $D_L$ in 15f.

## G.5  German credit data

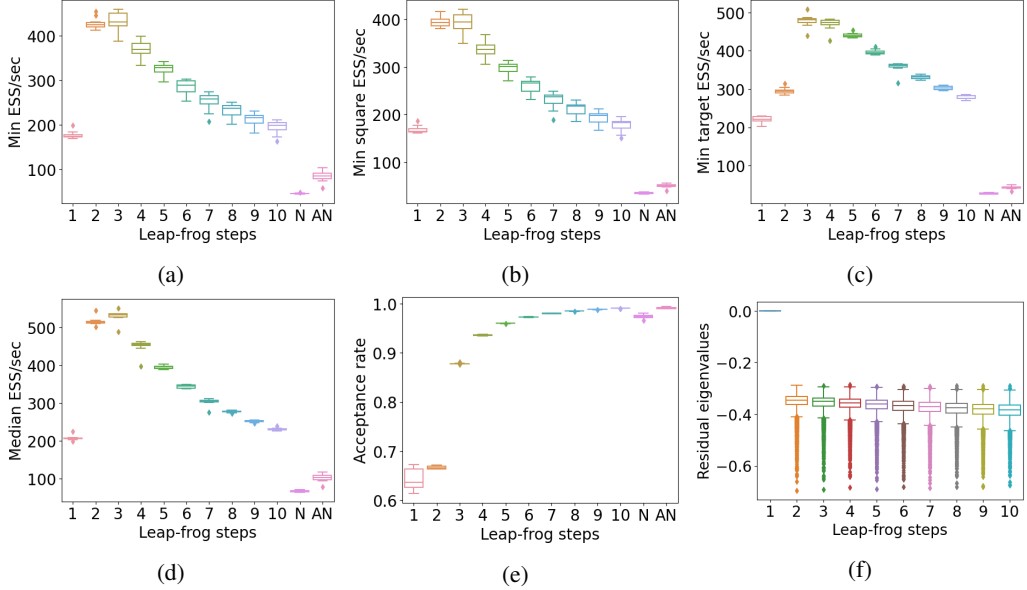

(a)  (b)  (c)

(d)  (e)  (f)

Figure 16: Bayesian logistic regression for German credit data set ($d = 25$). Minimum effective sample size per second after adaptation of $q \mapsto q_i$ in 16a, of $q \mapsto q_i^2$ in 16b and of $q \mapsto \log \pi(q)$ in 16c. Median marginal effective sample per second in 16d and average acceptance rates in 16e and estimates of the eigenvalues of $D_L$ in 16f.

## G.6 Caravan data

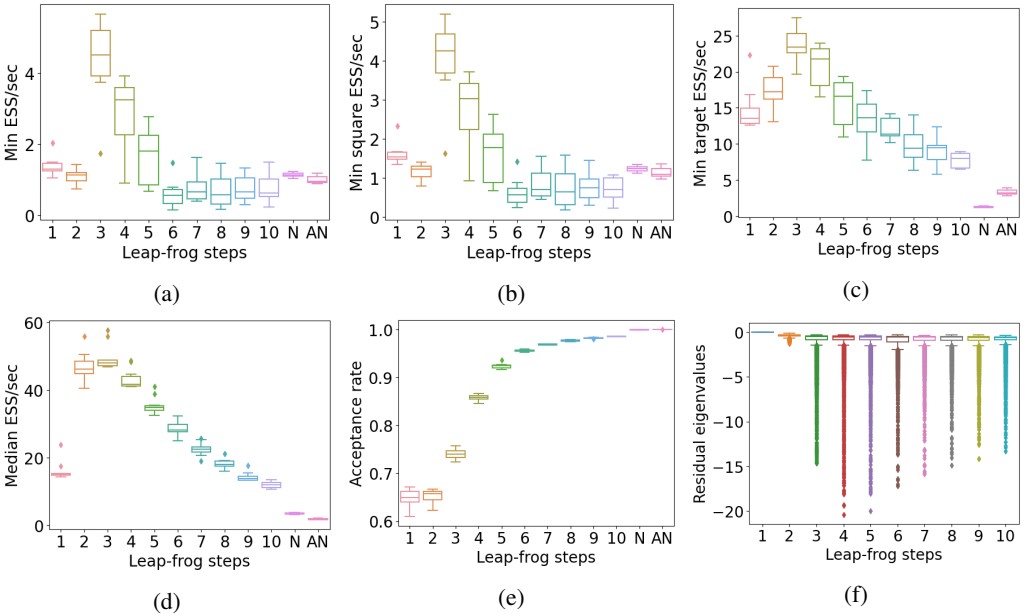

(a)  (b)  (c)

(d)  (e)  (f)

Figure 17: Bayesian logistic regression for Caravan data set ($d = 87$). Minimum effective sample size per second after adaptation of $q \mapsto q_i$ in 17a, of $q \mapsto q_i^2$ in 17b and of $q \mapsto \log \pi(q)$ in 17c. Median marginal effective sample per second in 17d and average acceptance rates in 17e and estimates of the eigenvalues of $D_L$ in 17f.

# H  Log-Gaussian Cox Point Process

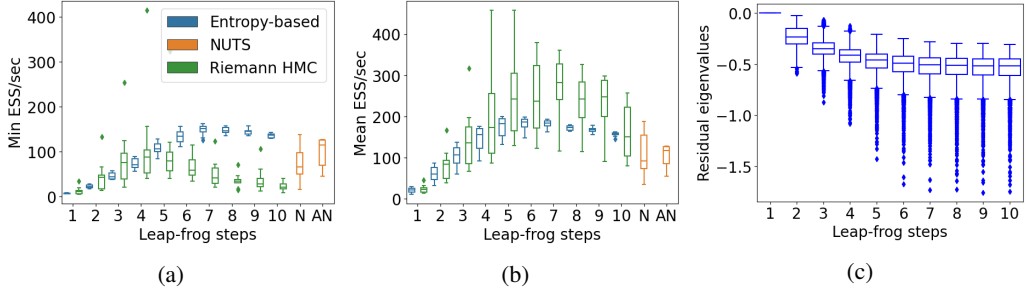

(a)  (b)  (c)

Figure 18: Cox process in dimension $d = 64$. Minimum (18a) and mean (18b) effective sample size per second after adaptation. Estimates of the eigenvalues of $D_L$ using power iteration in (18c).

# I  Stochastic volatility model

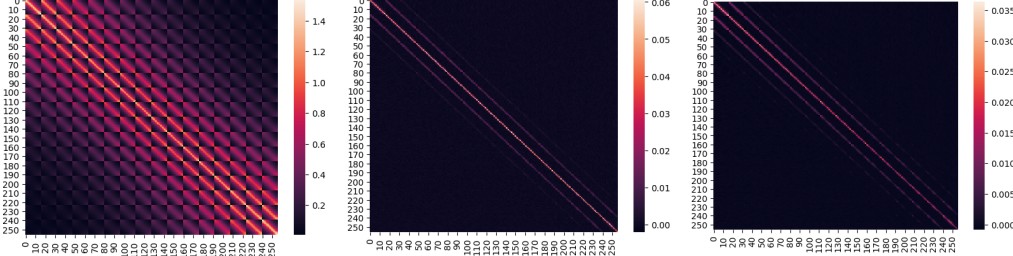

(a) Inverse mass matrix $(\Lambda +$
$\Sigma^{-1})^{-1}$ of the Riemann manifold based samplers.

(b) Inverse mass matrix $CC^\top$ for the entropy-based scheme with $L = 1$.

(c) Inverse mass matrix $CC^\top$ for the entropy-based scheme with $L = 5$.

Figure 19: Inverse mass matrices for the Cox process with $d = 256$ for the different schemes.

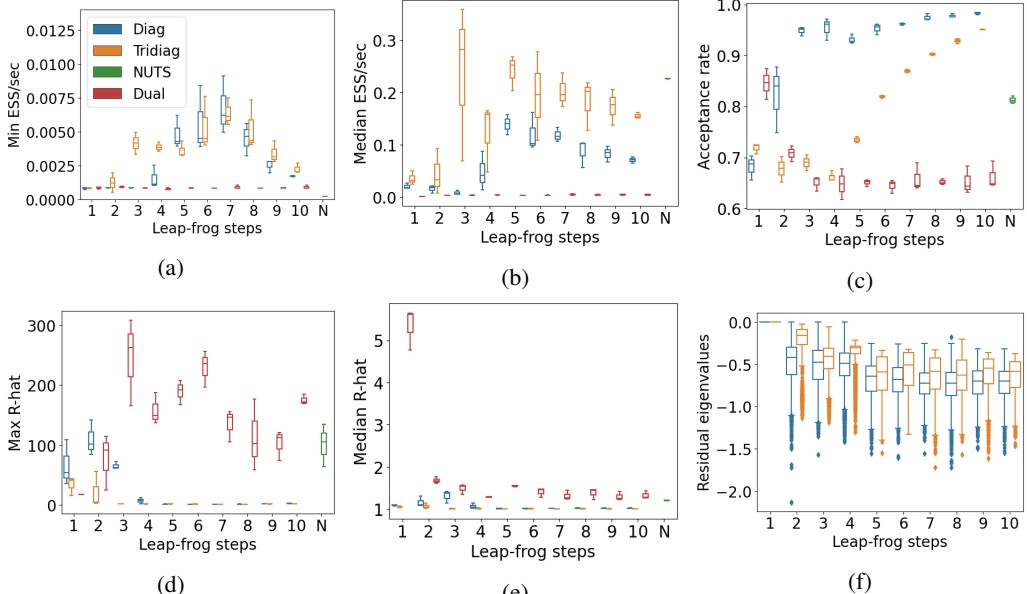

Figure 20: MCMC mixing efficiency for the stochastic volatility model ($d = 2519$) after adaptation: Minimum (20a) and median (20b) effective sample size per second. Maximum (20d) and median (20e) $\hat{R}$ of $q \mapsto q_i$. Average acceptance rates (20c) and estimates of the eigenvalues of $D_L$ (20f).



(a) First 100 dimensions of $M^{-1}$ for $L = 5$ with a tridiagonal mass matrix.

(b) Last 100 dimensions of $M^{-1}$ for $L = 5$ with a tridiagonal mass matrix.

(c) Last 100 dimensions of $M$ for $L = 5$ with a tridiagonal mass matrix.

Figure 21: Learned (inverse) mass matrices for the stochastic volatility model.