# OpenReview forum: "Entropy-based adaptive Hamiltonian Monte Carlo"
_NeurIPS.cc/2021/Conference — NeurIPS 2021 Poster_

### Official Review · Reviewer_URhX · 2021-06-30

**Rating:** 5
**Confidence:** 3

**Summary:**

In this paper, the authors propose an adaptive Hamiltonian Monte Carlo method. In their method, they adapt the hyperparameters of HMC by maximizing an approximation of the proposal entropy. And they show the efficiency of their method by experiments.

**Limitations And Societal Impact:**

There are some concerns and questions listed below. Since the reviewer is not familiar with HMC, there might exist some misunderstanding:

1. The reviewer is confused about (9) and the derivations before. There are two questions:

   [1]. From line 107-114, the authors try to derive a lower bound of $\log|\text{det}(1+DS)L(v)|$. And then the authors claim we can maximize the lower bound instead. However, if we already have a Gaussian target, then $DSL$ can be explicitly written down, why do we need to lower bound it? Even we want to do an extension later, the explicit formula can still be used. Can the authors give more comments on this?

   [2]. The equation (9) is really weird. If the target is not a Gaussian, then $\Sigma^{-1}$ should be replaced by summation of $\nabla^2U$. Then why it can be simply replaced by a single term of $\nabla^2U$. The intuition that follows seems unreasonable. Why can it be approximated by a Gaussian? Can the authors give more comments or detail on this?

2. In Section 3.2 line 151, the author claims that they solve (11) by lines 151-152. The reviewer thinks this is also a really big approximation. The authors basically replace an expectation with a one-particle estimation. Can the authors give more comments on this about why this is a reasonable approximation?

3. In Algorithm 1 line 10-line 13, the reviewer thinks the author runs one gradient descent for the minimization problem, is that right? If the authors only run one gradient descent, why does this give an (approximated) minimizer?  Besides, why do we need to project $\beta,\gamma$ on a compact set?

It seems that this paper has a lot of typos that affect reading. The reviewer lists some possible typos:

1. In line 84, the authors might make this $\theta$ clear. For example, write $C$ as $C(\theta)$. The reviewer feels it's hard to find the origin of this $\theta$ later.

2. In line 90-91 (3): $\mathcal{T}_L\rightarrow\mathcal{T}_i$ ?

3. Since there are typos in (3), the reviewers hope the authors can add calculations of (2) in the appendix.

4. In Lemma 1, what is $\|\cdot\|_p$?

5. In line 110, according to the lemma, we should have $\|C^\top\Sigma^{-1}C\|_2\leq \frac{h^2L^2}{4}$.

6. In Algorithm 1 line 5, what is (13)?



**Main Review:**

Originality, quality, significance: In this paper, the authors propose a new HMC method. Since the reviewer is not familiar with the HMC methods, the reviewer can't judge its novelty. However, the reviewer thinks the contribution is not enough. There are not any theoretical clues or justifications showing the efficiency of their new methods. And the reviewer also thinks that its intuition is not very reasonable.

Clarity: The reviewer thinks this paper is very unclear. There are a lot of typos.

Overall, the reviewer thinks that a paper that only contains empirical results is not good enough to be published in a theory-focus conference. The reviewer will list questions and concerns in "Limitation".

**Time Spent Reviewing:**

2.5 hours

---

> ### Author Response · Authors · 2021-08-10
> **Response to reviewer 3 (URhX)**
>
>
> We thank the reviewer for taking the time to review our submission and for the provided comments and feedback.
>
> We agree that the motivations and justifications for our approximations have been unclear at some points. This might have caused some misunderstandings and we try to be clearer in our revised version. We hope that the below help to clarify the concerns raised by the reviewer.
>
> Regarding concern 1:\
> Principally, one can compute and take gradients of $\log |\det (1+DS_L(v))|$ without any approximations using automatic differentiation, but this scales as $\mathcal{O}(d^3)$, which becomes impractical for many applications. Alternatively, one could principally also use the recursive formula that we have derived in Lemma 1. But note that for $L$ leapfrog steps, this requires computing $\frac{1}{2}L(L-1)$ terms, each involving the Hessian. This makes using the recursive formula computationally expensive even for a moderate number of leapfrog steps and using a stochastic trace estimator. For Gaussian targets, $DS_L$ can be decomposed into a term that contains all terms that are linear in $h^2C^\top \Sigma^{-1}C$ and that does not require a recursion; plus a term that contains terms that are higher than linear in $h^2C^\top \Sigma^{-1}C$ and that needs to be solved recursively. Our suggestion is to ignore this second term, as we expect it to be small because of the constraint that we have added to our objective.
> It is correct that for non-Gaussian targets, evaluating $DS_L$ exactly (recursively) leads to a non-linear function of the Hessians evaluated along the different points of the leapfrog-trajectory. Due to the computational reasons outlines above, we replace this polynomial with a first order term with one Hessian evaluation which is however scaled accordingly (notice the $L^2$ scaling) to account approximately for all the linear terms. This approximation will not be accurate if the Hessian varies significantly along the trajectory. Our experiments do however suggest that even such a crude approximation can be sufficient in practice.
>
> Regarding concerns 2 and 3:\
> It is indeed not clear that our optimization algorithm solves equation (11). Because we are interested in optimizing the mixing efficiency of the sampler at stationarity, the objective in (11) evaluates the speed measure under the target distribution, making it very challenging to optimize it exactly. We only aim to solve it approximately and we point this out more evidently in the revised version. In our implementation, we only perform one gradient step at each step of the Markov chain. It is of course possible to adjust this for example by performing multiple gradient steps before moving to the next state of the HMC chain, or only updating the parameters every say $k>1$ iterations. We expect that a good choice for the relative frequency of updating the pre-conditioning parameter versus the state of the HMC chain depends on initialisations for both the pre-conditioning parameters and the distribution of the initial HMC states. We have not explored this experimentally as the simple choice of a single gradient step seemed to perform well. We projected onto a compact set for numerical stability. For a relative high number of leapfrog steps, it can be optimal (in terms of the suggested objective) to have acceptance rates that are higher than those obtained from scaling arguments that assume a fixed integration time. Clipping the values of $\beta$ thus prevents it from diverging.
>
>
> It is possible to view the suggested adaptation scheme within a stochastic approximation framework of controlled Markov chains, see for instance references [1-3]. We have not presented it in this formality so as to not distract the readers from what we believe are more central points.
> We are indeed not aware of any work that has shown theoretically that any such adaptations for HMC for the targets considered in this paper do indeed yield converging ergodic averages. In light of this, we feel that our contributions are still of value for the NeurIPS community.
>
>
> We also want to address the typos or parts that were unclear:
>
> 1)	We thought a lot about whether we explicitly include $\theta$ in the notation of $C$, but eventually decided to just write $C$ because it felt less cluttered. However, we will try to better emphasis the dependence of $C$ on $\theta$.
>
> 2)	Yes, this is a typo. The subsequent developments should have the correct indices.
>
> 3)	We will add a proof of (3) for completeness. Analogous calculations for equation (2) in the case of an identity mass matrix can be found in the mentioned literature.
>
> 4)	$||\cdot||_p$ is just the standard p-norm. This generality is not really needed and we will replace it with the Euclidean norm which is used in our implementation.
>
> 5)	Yes, this is a typo, $h^2$ should also be in the denominator.
>
> 6)	Equation (13) can be found in the supplementary material. We will make this reference clearer.
>
> Again, we thank the reviewer for the constructive comments and questions. We believe that our manuscript will be much improved by incorporating these points.
>
> References:\
> [1] Andrieu and Thoms, A tutorial on adaptive MCMC, 2008.\
> [2] Andrieu and Moulines, On the ergodicity properties of some adaptive MCMC algorithms, 2006.\
> [3] Andrieu et al., On the stability of some controlled Markov chains and its applications to stochastic approximation with Markovian dynamic, 2015.

---

> > ### Comment · Reviewer_URhX · 2021-08-11
> > **Response to authors**
> >
> > The reviewer thanks the authors for the response.
> >
> > It seems that this paper proposes a method without theoretical justification and intuition for efficiency. The paper and the answer give the reviewer a feeling that all the methods and algorithms are designed randomly. And the method accidentally works in experiments.
> >
> > Thus, the reviewer hopes to see enough experiment results to support the efficiency of this method. Based on the authors' answer to Reviewer ZogC, the reviewer would like to raise the score to 5. And the review will wait for other reviewers to check if these experiments are good enough.

---

> > > ### Author Response · Authors · 2021-08-24
> > > **Thanks for the comments**
> > >
> > > We thank the reviewer for having had a look at our initial response to reviewer ZogC that detailed the disadvantages of possible gradient-based adaptation schemes using the ESJD for Gaussian targets and we are happy to hear that the reviewer would like to raise the initial score.
> > >
> > > In the subsequent responses to reviewer ZogC, we have now also included additional experiments for different targets (ill-conditioned Gaussians, logistic regression models, Log-Gaussian Cox process model) that indicate that the suggested approach can be competitive also with a more common NUTS setup that adapts a diagonal mass matrix. We hope that these results can provide at least some experimental evidence that the suggested adaptation scheme is not just working by accident.

---

### Official Review · Reviewer_K2Fq · 2021-07-16

**Rating:** 7
**Confidence:** 3

**Summary:**

The paper develops a gradient-based algorithm that enables adaptation of the mass matrix by taking into account the leapfrog integration error. The adaptation strategy incorporates an approximation of the proposal entropy in comparison to previous approaches that use other techniques such as the expected squared jumping distance.

**Main Review:**

# Originality:

My main concern reading the paper was whether the authors would be able to significantly add to the work of [45]. However, the more I went through the paper, the more I could see that there was a definite contribution and that this approach enabled the direct application of the generalized speed measure to a multiple step scenario in HMC.

Given the clear writing and excellent references to existing work, I am confident that the work is placed sufficiently among previous works.

# Quality and Clarity

This paper is written to a high standard and is structured well. I feel I was able to follow the main arguments displayed in the maths due to the thoughtful structuring of the theory section. However I am unable to verify and validate every equation (not due to the author’s writing style, but due to my finite time constraints!).

The Figures are a bit small, and perhaps the text could be a larger font size, as well as appropriately capitalising the axis labels. Furthermore, the experiments section could be a bit more focused on what the key take-aways are. I can see from the results that the adaptation works but I am not quite sure what exactly I am looking out for in particular. To me it seems previously known that a larger number of leap-frog steps leads to a better performance in higher dimensional settings. Perhaps more details of the overall take-away from the numerical experiments could be clarified further in the rebuttal? Finally, the experiment in Section 4.5 is very impressive and I actually think that the paper is too “humble” about it. Maybe it was referenced earlier in the paper, but I was surprised (in a good way) to see it pop up at the end!

Section 5 is well-written and demonstrates good knowledge of the area and limitations of the approach.

# Comments and Questions:
1)	How does the computational complexity of estimating the Hessian compare to just performing RMHMC directly? I am assuming that the technique is to use the Hutchinson stochastic trace estimation to approximate the Hessian using the vjp? I got a bit confused about the vector $w$, how does that get included?
2)	In the previous work of [45], it is clear that they are adapting the proposal distribution (e.g. the covariance). For HMC the proposal is defined using the Hamiltonian. I just wanted to check that the parameters that are being adapted is just the lower triangular $C$ of the mass matrix in the kinetic term?
3)	I actually had a question earlier on in the paper that was subsequently answered. However, I think it would be helpful to mention the need to only perform this adaptation during the burn-in stage a bit earlier in the paper.
4)	What is “stop_grad” in algorithm 1?
5)	Allowing NUTS to have a tree depth of $10$ leads to a maximum trajectory length of $2^{10}$. In my experience NUTS will often take advantage of the largest tree depth that is given to it (especially in high-dimensional spaces). Were other tree depths tried and is this why the Min ESS/sec is significantly lower for NUTS? In the paper the adaptive strategy only goes up to an $L=10$, maybe using a tree depth $2^5 = 32$ would be a fairer comparison?


### After Author Response:

I would like to thank the author(s) for their responses to all the reviewers (especially with the additional experiments in response to ZogC). As a result I am happy to keep my current score.

**Time Spent Reviewing:**

5

---

> ### Author Response · Authors · 2021-08-10
> **Response to reviewer 2 (K2Fq)**
>
>
> We thank the reviewer for taking the time to review our submission and for the provided comments and feedback. We appreciate the reviewer's positive reception.
>
> We agree that our experimental section has lacked some guidance as to what to take away from each experiment. A clearer outline of the contributions will be included in the updated version and we summarize them below:
>
>
> Our results with Gaussian targets show (i) that an adaptive version of HMC can scale better than MALA. As the reviewer has remarked, this is probably what we would expect from previous scaling arguments in HMC and it confirms this. The experiments also show (ii) that the adaptation scheme can learn a mass matrix that is adjusted to the geometry of the target. This can be seen from the fact that the eigenvalues of $C^\top \Sigma^{-1} C$ are close to one so that all dimensions of the target mix equally well. Our submitted version did not include experiments that indicate that this does often not hold if we replace the suggested objective with versions of the expected squared jumping distance, but see the response to reviewer 1 for more details that such alternative objectives can yield to efficient mixing in a few components only. We also compare the entropy-based adaptation with Riemann-Manifold based samplers for a Log-Gaussian Cox point process models. We find that both schemes mix similarly, which indicates that the gradient-based adaptation scheme can learn a suitable mass matrix without having access to the expected Fisher information matrix. Then, we consider a high-dimensional stochastic volatility model where the entropy-based scheme performs favourably compared to alternatives and illustrate that efficient sparsity assumptions can be accommodated when learning the mass matrix. Finally, we show in a toy example how the suggested approach might be modified to sample from highly non-convex potentials.
>
> We also like to address the further comments and questions raised:
>
> 1) The computational complexity of RMHMC is generally $\mathcal{O}(d^3)$ to sample from $d$-dimensional targets with $d>L$. The Log-Gaussian Cox point process is a special ideal case where the metric tensor becomes constant which yields a scaling of $\mathcal{O}(d^2)$. If we let $m$ be the expectation of $N$ (the number of terms in the Taylor series), then using the stochastic trace estimator scales as $\mathcal{O}(Ld^2+md^2)$ for a full Cholesky preconditioning matrix, whereas using a diagonal preconditioning matrix yields a scaling of $\mathcal{O}(Ld+md)$. In our experiments, $m$ was of order 5.
>
> 2)	Yes, we adapt just some matrix $C$ such that $CC^{\top}=M^{-1}$ is the inverse of the mass matrix. We can fix the step size $h$ as all terms involving $h$ and $C$ are of the form $h C$. Like in previous work, we considered $C$ to be a diagonal or a lower triangular matrix. However, as illustrated in the stochastic volatility model, we can also use alternative efficient parameterisations such as $C=B^{-1}$ where $B$ has non-zero entries only on the diagonal and the first upper off-diagonal. Since computations with $B$ are of order $d$, we can learn efficiently a HMC sampler where $C$ or the inverse mass matrix is dense, but the mass matrix itself is sparse.
>
> 3)	We will add the point that we perform adaptation only during burn-in earlier in the paper.
>
> 4)	The stop\_grad(y) operation means that we disregard any gradients (with respect to the parameters of $C$) in the computation of y. Note that we want to compute here a gradient over the Russian Roulette estimator and the straightforward approach would indeed be to just compute the gradient of all the terms without any gradient stopping. We have chosen here a slightly more efficient approach suggested in [1] that uses a Neumann series and requires only taking gradients of one term in the sum. The corresponding equation for this gradient expression is given in the first equation in the supplementary material.
>
> 5)	In our initial experiments, we used a maximum tree length of 10 for NUTS simply because this was the default setting in tensorflow probability, without giving much thought to it. Indeed, this leads to many leapfrog-steps being taken in the high-dimensional Gaussian targets. We have rerun the experiments using a maximum tree depth of 5 for some examples detailed in the response to reviewer 1, but the main message remains the same in that NUTS (without some adaptation of the mass matrix) mixes less efficiently.
>
> Again, we thank the reviewer for the the positive reception and the constructive feedback that we expect will improve our manuscript.
>
>
> References:\
> [1] Chen et al., Residual Flows for Invertible Generative Modeling, 2019.

---

### Official Review · Reviewer_ZogC · 2021-07-16

**Rating:** 6
**Confidence:** 4

**Summary:**

The paper proposes to adapt the mass matrix used in HMC by maximising the speed measure introduced in [1].
Estimating the speed measure is non-trivial and it is achieved by firstly finding a lower bound of the density induced by a HMC kernel and then estimating the bound using Russian-roulette sampling.
Experiments performed on multiple targets show that the method is working in practice.

[1] Michalis Titsias and Petros Dellaportas. Gradient-based adaptive Markov chain Monte Carlo. NeurIPS, 2019.

**Limitations And Societal Impact:**

Limitations are discussed in section 5.
No societal impact is discussed but it's not appleid to the paper.

**Main Review:**

Originality:
Successfully using the speed measure introduced in [1] to adapt mass matrix in HMC is novel contribution to the field.
Related work has been adequately cited and discussed.

Quality:
I appreciate the technical aspects of the method which successfully adapts the mass matrix using the speed measure from [1] via gradients,
but I found the experiments are poorly done.
In particular, the main claim of the paper is that the speed measure from [1] is a better metric to adapt in compare to expected squared jumping distance (ESJD) (or other metrics).
However, no direct comparisons to alternative metrics are done but only NUTS, a dynamics HMC method, and (in some experiments) Riemann HMC are compared.
The author(s) should compare normal HMC with mass matrix adapted using ESJD, e.g. those methods discussed in the related work section.
Besides, the setup of NUTS is weird.
- Why do you choose NUTS with only step size adaptation while comparing to the proposed method with mass matrix adaptation? Even mass matrix is adapted, NUTS is not using ESJD as a metric for adaptation.
- Why do you use a maximum depth of $10$? Shouldn't you use a more comparable value as to $L$ for the proposed method? I suspect figure 1 can change a lot if the parameter of NUTS is set differently.

Clarity:
The method itself is easy to follow but I found the experiments section is poorly done, as I commented above.
Experiments that are designed to support the main claim are missing and the current ones tend to evaluate some characteristics of the proposed method, which are interesting but not of the top priority, e.g. why all figures focus on evaluating how ESS changes with different number of leapfrog steps?

Significance:
The method is technically involved but the current results are limited to convince practical use of the method.

Misc:
- I quite like the outlook discussed in section 5. One thing I would add is that adapting the proposed method to multinomial HMC can be interesting as the simulation error is less of a problem there. It could make the point on coupled HMC in section 5 more interesting too, as [2] recently shows that multinomial HMC is more effective than (standard) HMC in terms of couplings.
- You should capitalise words like **M**arkov in your references. For `bibtex` used in $\LaTeX$, you can achieve so by adding `{}` to the letter to make sure it's capital.

[1] Michalis Titsias and Petros Dellaportas. Gradient-based adaptive Markov chain Monte Carlo. NeurIPS, 2019.

[2] Kai Xu, Tor Erlend Fjelde, Charles Sutton, Hong Ge. Couplings for Multinomial Hamiltonian Monte Carlo. AISTATS, 2021

---

Update after new results (other alternative metrics and NUTS):

- The comparisons against ESJD and L2HMC support the motivation of using the speed measure as the optimisation metric, which is crucial.
- The comparisons against a more common setup of NUTS (dual-averaging + diagonal mass matrix adaptation) is useful to show that the proposed methods can outperform the "standard" NUTS with more than one hyper-parameter setup (which is important to see the method is not sensitive to a particular one) and supports the empirical usefulness of the method.

I encourage the author to release the source code of the method as a ready-to-use TFP add-on as the method itself can be a bit involved to implement.

**Time Spent Reviewing:**

2.5

---

> ### Author Response · Authors · 2021-08-10
> **Response to reviewer 1(ZogC)**
>
> We thank the reviewer for taking the time to review our manuscript and for the provided comments and feedback.
>
> We very much appreciate the reviewer's positive reception for the technical aspects of our  submission. We agree with the point made by the reviewer that we did not compare our adaptation scheme with a normal (i.e. non-dynamic/NUTS) adaptation approach for the mass matrix that uses the expected squared jumping distance (ESJD). The stopping criterion of NUTS is motivated by the ESJD; it can be seen as stopping at a stationary point of the ESJD, in the hope that it is a maximum, see for example reference [1, Appendix D]. Because NUTS as published (i) does not adapt the mass matrix and (ii) is inherently dynamic in terms of the number of leapfrog steps, we agree with the reviewer that comparing with NUTS is not the most natural benchmark to show the efficiency of the speed measure objective. Nevertheless, since NUTS is a commonly used state-of-the art HMC sampler, we believe that a method can be of practical use if it mixes more efficiently compared to NUTS.
>
>
> To address the reviewer's concern about a lack of comparison using a direct optimisation of the mass matrix based on the ESJD, we have run additional experiments for example for the Gaussian targets in our submission. Two objectives have been considered as a replacement of the generalised speed measure (GSM):
> (a) simply the ESJD;
> (b) a weighted combination of the ESJD and its inverse as suggested in [2], without any burn-in component, which we denote L2HMC.
>
> For the ill-conditioned anisotropic Gaussian with $d=100$, the minESS/sec results are as follows (where we report the median and standard deviation in parenthesis using 10 replications):
>
> | Steps                          | GSM       | ESJD| L2HMC | NUTS|
> |---------------------------------|-------------|---------|-----------|------|
> | L=1                     | 122.34  (15.53)   |  0.10  (0.01)     |  0.10   (0.01)         | NA |
> | L=5                     | 753.76  (22.23)   |   0.09  (0.02)     | 0.09  & (0.02)         | NA |
> | L=10                     | 569.98  (37.41)   |   0.59  (395.20)    |  0.14  & (0.05)        | NA |
> | max tree depth of 5                     | NA   |  NA     | NA         |  0.05 & (0.01) |
> | max tree depth of 10                     | NA   |  NA    |  NA         |  0.56 & (0.04) |
>
> We have also computed the meanESS/sec (across the dimensions):
>
> | Steps                          | GSM       | ESJD| L2HMC | NUTS|
> |---------------------------------|-------------|---------|-----------|------|
> | L=1                     | 171.59  (7.34)   | 72.44   (1.13)     |  65.26   (1.76)         | NA |
> | L=5                     | 796.94  (20.48)   |   479.96  (39.80)     | 481.37  (38.72)        | NA |
> | L=10                     | 605.38  (32.40)   |   927.57  (17.60)   |  739.81  (48.38)       | NA |
> | max tree depth of 5                     | NA   |  NA     | NA         |  37.94  (1.35) |
> | max tree depth of 10                     | NA   |  NA    |  NA         | 3.10  (0.15) |
>
>
>
> It can be observed that while the objectives based on the ESJD can yield higher average ESS/sec for L=10, this is accompanied with a very small ESS/sec for some components. This effect also becomes evident if one looks at the condition number of $C^\top \Sigma^{-1}C$. Figure 8 in our submission shows that for the GSM objective, this condition number is close to one. Conversely, its median is greater than $10^3$ for L2HMC and greater than $10^4$ for ESJD for all leapfrog steps ranging form 1 to 10.
> The fact that the ESJD can be high even when some components mix poorly has been stated as a major motivation for the GSM introduced in [3], but it has not been demonstrated explicitly therein and we hope that this provides some empirical evidence for this.
>
> The above results also include results for NUTS with a maximum tree length of 5 which seem to not improve the mixing speed significantly. We are thankful to the reviewer for pointing out that our previous choice for a maximum tree depth of 10 is not well motivated. We have used the NUTS implementation in tensorflow probability where this choice was the default value, but would appreciate any feedback from the reviewer if the adjusted NUTS setup still feels weird.
>
> We would like to mention that optimizing the ESJD can lead to poor mixing also if the variances of each component are similar. Indeed, the minESS/sec statistics for the correlated Gaussian target from our submission are as follows:
>
> | Steps                          | GSM       | ESJD| L2HMC | NUTS|
> |---------------------------------|-------------|---------|-----------|------|
> | L=1                     | 63.84   (3.93)   |  0.75  (1.57)    |  0.25   (0.12)        | NA |
> | L=5                     |  389.96  (5.02)  |   2.01  (5.37)     | 2.65   (2.33)        | NA |
> | L=10                     | 282.66 (7.81)  |  0.92  (3.70)   |  0.36   (0.85)        | NA |
> | max tree depth of 5                     | NA   |  NA     | NA         |  4.09   (0.34)|
> | max tree depth of 10                     | NA   |  NA    |  NA         | 12.49  (1.80)|
>
> while the mean ESS/sec are below:
>
> | Steps                          | GSM       | ESJD| L2HMC | NUTS|
> |---------------------------------|-------------|---------|-----------|------|
> | L=1                     | 71.70   (4.47)   |   7.98   (21.67)    |  1.18   (1.14)     | NA |
> | L=5                     |  413.73  (4.19)  |   36.26  (49.87)     |12.30  (8.86)       | NA |
> | L=10                     | 298.66  (6.30)  |  22.64  (63.35)  |  4.28   (110.69)       | NA |
> | max tree depth of 5                     | NA   |  NA     | NA         |  4.66   (0.39) |
> | max tree depth of 10                     | NA   |  NA    |  NA         | 13.37  (1.87) |
>
>
>
> Instead of using the ESJD, it is possible to weight the distances differently, for example based on the inverse of the sample covariance matrix, but this scales as $\mathcal{O}(d^3)$ in general. Lastly, we just want to clarify that we simply plot the ESS statistics for different number of leapfrog steps to illustrate that our method is not overly sensitive to the number of leapfrog steps.
>
> Again, we thank the reviewer for the suggestions. We have not thought about including a more direct benchmark previously and expect that the additional experiments will lead to an improved presentation of our work.
>
> References:\
> [1] Andrieu et al., A general perspective on the Metropolis-Hastings kernel, 2020.\
> [2] Levy et al., Generalizing Hamiltonian Monte Carlo with Neural Networks, 2018.\
> [3] Titsias and Dellaportas, Gradient-based Adaptive Markov Chain Monte Carlo, 2019.

---

> > ### Comment · Reviewer_ZogC · 2021-08-16
> > **Thanks for the author response**
> >
> > > Because NUTS as published (i) does not adapt the mass matrix and (ii) is inherently dynamic in terms of the number of leapfrog steps, we agree with the reviewer that comparing with NUTS is not the most natural benchmark to show the efficiency of the speed measure objective. Nevertheless, since NUTS is a commonly used state-of-the art HMC sampler, we believe that a method can be of practical use if it mixes more efficiently compared to NUTS.
> >
> > Although the original NUTS paper doesn’t include mass matrix adaptation, it’s available in almost all modern PPLs (Stan, PyMC3, Turing, TFP, etc) as the standard NUTS to use nowadays.
> > So if the goal is to check if the proposed method would be preferred in practice, the “modern NUTS” [1] is the one that should be compared against.
> >
> > More importantly, it is the way to adapt the mass matrix to be compared here.
> > Modern NUTS simply uses a naive heuristics: The mass matrix is computed based on previous sample history, rather than maximising any metric.
> > So I’d expect the proposed method to be better.
> > Also I believe it’s possible to construct either normal HMC or NUTS with the mass matrix adaption in TensorFlow Probability or AdvancedHMC.jl.
> >
> > > To address the reviewer's concern about a lack of comparison using a direct optimisation of the mass matrix based on the ESJD, we have run additional experiments for example for the Gaussian targets in our submission.
> >
> > Thanks for the additional experiments.
> > Now the differences between GSM vs ESJD/L2HMC give me more confidence on the method (I quite like the differences between min and mean + your explanation on why it is the case) but I still have a few concerns.
> >
> > 1. If I understand correctly, the new results correspond to figure 8 in the appendix. Why do we switch the focus of figure 1 in the main paper (anisotropic Gaussian) to this setting now?
> >   - Does the same trend hold for anisotropic Gaussian?
> >   - Does the same trend hold for more complex distribution other than Gaussians? For me to raise my score, I need to see other targets beyond Gaussians.
> > 2. For $L=10$ in the first table, the standard deviation for ESJD is very large - Can you tell me how the 10 replicates look like?
> >
> > > The above results also include results for NUTS with a maximum tree length of 5 which seem to not improve the mixing speed significantly. We are thankful to the reviewer for pointing out that our previous choice for a maximum tree depth of 10 is not well motivated. We have used the NUTS implementation in tensorflow probability where this choice was the default value, but would appreciate any feedback from the reviewer if the adjusted NUTS setup still feels weird.
> >
> > Still, I think you should compare to TFP’s NUTS with mass matrix adaptation on as well. In terms of max depth, can you also vary it a bit more? A depth of 5 means $L=2^5$, which is still not comparable to other settings.
> >
> > [1] Michael Betancourt, A Conceptual Introduction to Hamiltonian Monte Carlo, 2017

---

> > > ### Author Response · Authors · 2021-08-18
> > > **Comparison with NUTS using an adapted mass matrix**
> > >
> > > We thank the reviewer for the useful comments regarding our additional experiments!
> > >
> > > We have followed the suggestion to also adapt a diagonal mass matrix for a NUTS implementation in order to have a clearer comparison between the different adaptation schemes.
> > >
> > > The results for the less ill-conditioned anisotropic Gaussian target ($d=1000$) from Figure 1 are as follows for the MinESS/sec:
> > >
> > > | Steps                          | GSM       | ESJD| L2HMC | NUTS|
> > > |---------------------------------|-------------|---------|-----------|------|
> > > | L=1                     |  13.53 	(5.54) |  0.01 	(0.02)    |   0.01 	(0.02)         | NA |
> > > | L=5                     |  159.34 	(58.42)   |    0.01 	(0.02)   | 0.03 	(0.02)          | NA |
> > > | L=10                     | 194.82 	(63.60)  |   0.01 	(0.02)  |   0.04 	(0.03)     | NA |
> > > | adapted step-size                      | NA   |  NA     | NA         |  18.36 	(3.54)|
> > > | adapted mass matrix                  | NA   |  NA    |  NA         |  48.83 	(16.94) ​|
> > >
> > > We have not included the meanESS/sec values in our initial submission, but the computed values below are from the same MCMC outputs used to generate Figure 1 for the GSM objective:
> > >
> > > | Steps                          | GSM       | ESJD| L2HMC | NUTS|
> > > |---------------------------------|-------------|---------|-----------|------|
> > > | L=1                     |   21.61 	(8.32) |   2.48 	(3.40)   |    2.21 	(4.69)        | NA |
> > > | L=5                     |   167.32 	(61.28)   |     21.44 	(35.85) | 62.73 (47.28) | NA |
> > > | L=10                     |  201.93 	(66.20)  |    106.78 	(79.89) |    100.22 	(85.66)    | NA |
> > > | adapted step-size                      | NA   |  NA     | NA         |   19.66 	(3.41)|
> > > | adapted mass matrix                  | NA   |  NA    |  NA         |   50.29 	(17.43) ​|
> > >
> > > One can observe that NUTS with adapted mass matrix mixes here more efficiently compared to the MALA case $L=1$, but appears less efficient compared to $L=5$ or $L=10$. For additional context, NUTS with adapted mass matrix most often used 7 leapfrog steps. The difference between min and mean ESS for the gradient-based adaption schemes using the ESJD or L2HMC objective is also evident here.
> > >
> > > The results for the very ill-conditioned Gaussian ($d=100$) from Figure 8 are below, now also including the mass matrix adaptation for NUTS:
> > >
> > > | Steps                          | GSM       | ESJD| L2HMC | NUTS|
> > > |---------------------------------|-------------|---------|-----------|------|
> > > | L=1                     | 122.34  (15.53)   |  0.10  (0.01)     |  0.10   (0.01)         | NA |
> > > | L=5                     | 753.76  (22.23)   |   0.09  (0.02)     | 0.09   (0.02)         | NA |
> > > | L=10                     | 569.98  (37.41)   |   0.59  (395.20)    |  0.14   (0.05)        | NA |
> > > | adapted step-size                      | NA   |  NA     | NA         |  0.05  (0.01) |
> > > | adapted mass matrix                  | NA   |  NA    |  NA         |  287.13  (85.94) ​|
> > >
> > >
> > > with the MeanESS/sec being
> > >
> > > | Steps                          | GSM       | ESJD| L2HMC | NUTS|
> > > |---------------------------------|-------------|---------|-----------|------|
> > > | L=1                     | 171.59  (7.34)   | 72.44   (1.13)     |  65.26   (1.76)         | NA |
> > > | L=5                     | 796.94  (20.48)   |   479.96  (39.80)     | 481.37  (38.72)        | NA |
> > > | L=10                     | 605.38  (32.40)   |   927.57  (17.60)   |  739.81  (48.38)       | NA |
> > > | adapted step-size                     | NA   |  NA     | NA         |  37.94  (1.35) |
> > > | adapted mass matrix                     | NA   |  NA    |  NA         | 287.13	(85.56)
> > >  |
> > >
> > > We have chosen to report this example (and not Figure 1) in the previous reply to illustrate that using a gradient based optimization for the ESJD or L2HMC objective can lead to very high meanESS/sec values, although some components have a very low ESS/sec. Note that just looking at the meanESS would suggest that these methods outperform NUTS with an adapted mass matrix, but for NUTS with adapted mass matrix, meanESS and minESS are extremely close.
> > >
> > > As you have noticed, the standard deviation for the ESJD objective can be high for the minESS statistic. This was caused by one replication that managed to learn a mass matrix that is very aligned to the target (the condition number of the transformed Hessian became well-conditioned as for the GSM objective). We are not so sure why this happened once, all replications differ by a random seed that affects (i) the initial positions and (ii) the initial pre-conditioning parameters by a small perturbation. Generally, we found in our implementation that both the ESJD and L2HMC objectives require also some tuning in terms of learning rates and gradient clipping and more MCMC steps (compared to the GSM objective) tend to be required before reaching a stationary point for the pre-conditioning parameters, particularly for learning a dense mass matrix using a Cholesky factor.
> > >
> > > The mass matrix for NUTS has been adapted using tfp.experimental.mcmc.windowed_adaptive_nuts.
> > > We are not aware of a method to adapt a dense mass matrix in the official tensorflow probability library. In our updated version of the paper, we will however try to be clearer that adaptation for a diagonal and/or full mass matrix within NUTS is indeed available in different PPLs.
> > >
> > > We have also tried the same diagonal mass matrix adaptation scheme for NUTS with the logistic regression models. In our initial experiments, this did not improve on NUTS with just the step size adapted. We will now check this again before reporting the results. However, we found there again that the entropy objective improves on the ESJD objectives, assuming the same gradient-based optimization.

---

> > > > ### Author Response · Authors · 2021-08-18
> > > > **NUTS using an adapted mass matrix for logistic regression models and Log-Gaussian Cox process model**
> > > >
> > > > We have run additional experiments for a NUTS sampler that adapts a diagonal mass matrix for the logistic regression and the Cox process experiments that correspond to Figures 2-4. These are based on the tensorflow probability library using tfp.experimental.mcmc.DiagonalMassMatrixAdaptation and a dual-averaging adaptation of the step-size implemented therein.
> > > >
> > > > MinESS/sec for the German credit data set (logistic regression) corresponding to Figure 2:
> > > >
> > > > | Steps                          | GSM       | ESJD| L2HMC | NUTS|
> > > > |---------------------------------|-------------|---------|-----------|------|
> > > > | L=1                     |  173.46 	(8.86)   | 0.74 	(0.28)    |   0.70 	(0.57)        | NA |
> > > > | L=5                     |  327.65 	(15.19)  |    0.51 	(0.18)   |  0.51 	(0.13)    | NA |
> > > > | L=10                     | 198.74 	(15.93) |   0.67 	(0.69)   |   0.31 	(0.03)     | NA |
> > > > | adapted step-size                     | NA   |  NA     | NA         |  52.63 	(5.51)|
> > > > | adapted mass matrix                     | NA   |  NA    |  NA         |  85.25 	(12.98)
> > > >  |
> > > >
> > > > MeanESS/sec for the German credit data set:
> > > >
> > > > | Steps                          | GSM       | ESJD| L2HMC | NUTS|
> > > > |---------------------------------|-------------|---------|-----------|------|
> > > > | L=1                     |   205.18 	(6.49)   | 17.10 	(7.48)   |     20.19 	(7.28)     | NA |
> > > > | L=5                     |   393.04 	(3.54) |     174.03 	(44.98) |   181.43 	(49.34) | NA |
> > > > | L=10                     |  229.68 	(2.07)|     129.13 	(27.33)  |    134.04 	(29.06)   | NA |
> > > > | adapted step-size                     | NA   |  NA     | NA         |   79.29 	(8.83)
> > > > | adapted mass matrix                     | NA   |  NA    |  NA         |   99.46 	(11.76)
> > > >
> > > >  MinESS/sec for the Caravan data set (logistic regression) corresponding to Figure 3:
> > > >
> > > > | Steps                          | GSM       |NUTS|
> > > > |---------------------------------|-------------|------|
> > > > | L=1                     |   1.29 	(0.26)      | NA |
> > > > | L=5                     |   1.80 	(0.77)    | NA |
> > > > | L=10                     |  0.63 	(0.39)  | NA |
> > > > | adapted step-size                     | NA          |   0.76 	(0.76)|
> > > > | adapted mass matrix                     | NA           |   0.95 	(1.00)
> > > >
> > > >  MeanESS/sec for the Caravan data set:
> > > >
> > > > | Steps                          | GSM       |NUTS|
> > > > |---------------------------------|-------------|------|
> > > > | L=1                     |    14.92 	(3.04)     | NA |
> > > > | L=5                     |    29.77 	(2.26)  | NA |
> > > > | L=10                     |   11.33 	(0.74)| NA |
> > > > | adapted step-size                     | NA          |   8.43 	(0.33)|
> > > > | adapted mass matrix                     | NA           |    1.89 	(0.19)
> > > >
> > > > We had numerical issues optimizing the ESJD objective even for a very low learning rate. Optimizing the L2HMC objective was possible, but the results are not competitive.
> > > >
> > > > MinESS/sec for the Cox process model ($d=256$) corresponding to Figure 4:
> > > >
> > > > | Steps                          | GSM       | RHMC| NUTS|
> > > > |---------------------------------|-------------|---------|------|
> > > > | L=1                     |    2.64 	(0.70)  |  4.65 	(2.91)       | NA |
> > > > | L=5                     |    17.53 	(9.38)|     26.73 	(20.16)  | NA |
> > > > | L=10                     |   20.41 	(8.22)|      10.82 	(6.55)  | NA |
> > > > | adapted step-size                     | NA   |  NA         |    13.39 	(5.30)
> > > > | adapted mass matrix                     | NA   |  NA    |    15.71 	(8.59)
> > > >
> > > > MeanESS/sec for the Cox process model ($d=256$):
> > > >
> > > > | Steps                          | GSM       | RHMC| NUTS|
> > > > |---------------------------------|-------------|---------|------|
> > > > | L=1                     |     6.04 	(2.06)  |   6.71 	(3.76)     | NA |
> > > > | L=5                     |     39.96 	(14.07)|     57.11 	(30.57) | NA |
> > > > | L=10                     |    33.96 	(7.99)|      44.47 	(17.66)| NA |
> > > > | adapted step-size                     | NA   |  NA         |     26.81 	(6.63)
> > > > | adapted mass matrix                     | NA   |  NA    |     23.93 	(6.78)

---

> > > > > ### Comment · Reviewer_ZogC · 2021-08-24
> > > > > **Thanks again for the author response**
> > > > >
> > > > > It's nice to see the updated results based on a more common setup of NUTS (dual-averaging + diagonal mass matrix adaptation).
> > > > > I think the results now show that the proposed methods can outperform the "standard" NUTS with more than one hyper-parameter setup (which is important to see the method is not sensitive to a particular one) and supports the empirical usefulness of the method.
> > > > > Of course the comparisons against ESJD and L2HMC support the motivation of using the speed measure as the optimisation metric, which is crucial.
> > > > > I'm therefore happy to raise my score to 6 based on these new results.
> > > > > I encourage the author to release the source code of the method as a ready-to-use TFP add-on as the method itself can be a bit involved to implement.

---

> > > > > > ### Author Response · Authors · 2021-08-24
> > > > > > **Thanks for going through our additional results and the feedback**
> > > > > >
> > > > > > We thank the reviewer for taking the time to go through our additional experiments and raising the score to 6.
> > > > > >
> > > > > > The additional comparisons with both the ESJD/L2HMC objectives as well as the more common NUTS setup (dual-averaging + diagonal mass matrix) will be added to the experiments in the revised paper. We will release source code so that the suggested adaptation scheme can be used readily as a possible alternative to the adaptive MCMC kernels within TFP.

---

### Decision · Program_Chairs · 2021-09-27

**Decision:**

Accept (Poster)

**Comment:**

The paper introduces a gradient-based sampling algorithm with adaptive mass matrix to account for the integration error. The referees find the writing of the paper excellent and also agree upon the numerical experiments being convincing after multiple round of discussions. The paper should be accepted to the conference as a poster.